# Estimating Interventional Distributions with Uncertain Causal Graphs through Meta-Learning

**Anish Dhir**[*]
Imperial College London

**Cristiana Diaconu**[*]
University of Cambridge

**Valentinian Mihai Lungu**
University of Cambridge

**James Requeima**
University of Toronto
Vector Institute

**Richard E. Turner**
University of Cambridge
Alan Turing Institute

**Mark van der Wilk**
University of Oxford

## Abstract

In scientific domains—from biology to the social sciences—many questions boil down to *What effect will we observe if we intervene on a particular variable?* If the causal relationships (e.g. a causal graph) are known, it is possible to estimate the intervention distributions. In the absence of this domain knowledge, the causal structure must be discovered from the available observational data. However, observational data are often compatible with multiple causal graphs, making methods that commit to a single structure prone to overconfidence. A principled way to manage this structural uncertainty is via Bayesian inference, which averages over a posterior distribution on possible causal structures and functional mechanisms. Unfortunately, the number of causal structures grows super-exponentially with the number of nodes in the graph, making computations intractable. We propose to circumvent these challenges by using meta-learning to create an end-to-end model: the Model-Averaged Causal Estimation Transformer Neural Process (MACE-TNP). The model is trained to predict the Bayesian model-averaged interventional posterior distribution, and its end-to-end nature bypasses the need for expensive calculations. Empirically, we demonstrate that MACE-TNP outperforms strong Bayesian baselines. Our work establishes meta-learning as a flexible and scalable paradigm for approximating complex Bayesian causal inference, that can be scaled to increasingly challenging settings in the future.

## 1   Introduction

Answering interventional questions such as: "What happens to Y when we change X?" is central to areas such as healthcare [37] and economics [57]. One can estimate such *interventional distributions* by actively intervening on the variable of interest and observing the effects (obtaining *interventional data*), but this can be costly, difficult, unethical, or even impossible in practice [35]. Causal inference offers an alternative by leveraging readily available *observational data* alongside knowledge of the underlying causal relationships in the form of a causal graph [50]. A causal graph can be manually specified when domain knowledge is available. In the absence of this, causal discovery techniques attempt to learn the causal structure from data [45]. However, causal discovery from purely observational data is notoriously difficult. Identifying the true graph requires strong assumptions, such as the use of certain restricted model classes [54, 13, Ch. 4] and the acquisition of infinite data, that are rarely met in practice. It is therefore often the case that data provides plausible evidence for a set of causal graphs, even though each of these graphs may imply drastically different causal

---

[*]Equal contribution

39th Conference on Neural Information Processing Systems (NeurIPS 2025).

effects. Picking a single graph can thus result in poor downstream decisions [51, 4]. In this work, we address the challenge of tractably estimating interventional distributions when the true causal graph is uncertain, a common scenario in real world applications.

Instead of using a single graph to drive decisions, the Bayesian framework provides a principled way to manage the uncertainty over the causal models by maintaining a distribution over possible causal structures and functions relating variables. With access to both, model uncertainty can be accounted for by marginalising the interventional distributions over both posteriors (fig. 1) [43, 5, 40]. However, this procedure has two main challenges. First, the space of causal graphs grows super-exponentially with the number of variables, making exact posterior inference intractable, and sampling difficult to scale. Second, even with a posterior over causal graphs, estimating an interventional distribution in each plausible causal graph necessitates computing the posterior over functional mechanisms, which is analytically intractable except in simple models. Poor approximations at any point in this pipeline can result in inaccurate interventional distributions. Consequently, most works restrict themselves to simple functional mechanisms and constrain the allowable structures, limiting their applicability.

To overcome these bottlenecks, we turn to recent advances in meta-learning. Neural processes (NPs) [19, 20] are a family of meta-learning models that approximate the Bayesian posterior with guarantees [25], by *directly* mapping from datasets to the predictive distribution of interest, thus bypassing the intractable explicit modelling of intermediate posteriors. They underpin several successful methods with strong empirical performance across a range of real-life domains, ranging from tabular data classification [28] to weather modelling [63]. Recently, NPs have also been applied to causal discovery [30, 39, 14], with Dhir et al. [15] showing that they can accurately recover posterior distributions over causal graphs. However, these existing approaches cannot estimate interventional distributions.

In this work, we apply NPs to the problem of causal inference directly from data by developing a meta-learning framework that targets Bayesian posteriors for interventional queries—the Model-Averaged Causal Estimation Transformer Neural Process (MACE-TNP). As shown in fig. 1, our method amortises the full Bayesian causal inference pipeline (learning the posterior over the causal structure and functions, and marginalising over them), all within a single model. By directly estimating the interventional distribution, our approach avoids compounding errors from intermediate approximations of posteriors and marginalisation, enabling more accurate and computationally-efficient inference under model uncertainty. Our contributions are threefold. First, we propose an end-to-end model trained on synthetic datasets to directly approximate the Bayesian posterior interventional distribution. Second, we empirically show that when the analytical posterior is available in closed-form, MACE-TNP converges to it. Third, we demonstrate that MACE-TNP outperforms a range of Bayesian and non-Bayesian baselines across diverse experimental settings of increasing complexity, highlighting the method's potential to scale to high-dimensional datasets. Our framework paves the way for meta-learning-based foundation models for causal interventional estimation.

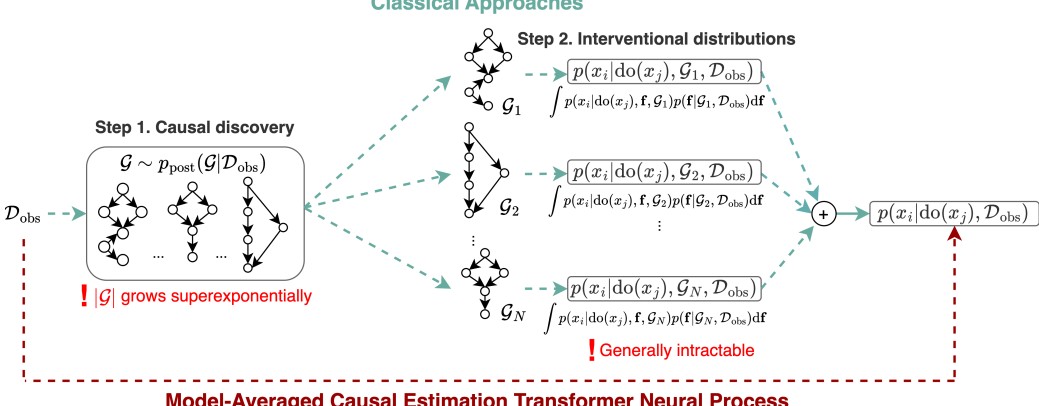

Figure 1: Overview of MACE-TNP. Unlike classical approaches, that usually require a two-step procedure which 1) first involves posterior inference over the graph structure, followed by 2) complicated inference over the functional mechanism, MACE-TNP amortises the full causal inference pipeline.

## 2 Background

Our goal in this paper is to compute interventional distributions that take uncertainty over the causal model—structure and functional mechanism—into account. Like a majority of works that learn causal models from data, we assume no hidden confounders. We set up the problem and provide background here. Throughout the paper random variables are denoted by uppercase letters (e.g., $X$), and their realizations by lowercase letters (e.g., $x$). Boldface is used to denote vectors (e.g., $\mathbf{x}, \mathbf{X}$).

**Causal model:** Causal concepts can be formally defined by considering a causal model. Given a *directed acyclic graph* (DAG) $\mathcal{G}$ with node set $V := \{1, \ldots, D\}$, functional mechanisms $\mathbf{f} := \{f_1, \ldots, f_D\}$, and independent noise terms $U := \{U_1, \ldots, U_D\}$, a *Structural Causal Model* (SCM) defines variables $X_i$ recursively as follows [50]

$$X_i := f_i(\mathrm{PA}_i, U_i), \qquad \text{for } i = 1, \ldots, D, \qquad (1)$$

where $\mathrm{PA}_i \subset \{X_1, \ldots, X_D\} \backslash \{X_i\}$ is the set of parents for $X_i$. This process induces a joint distribution over all the variables. Such a construction can then be used to formally define interventions. We focus on *hard interventions*, denoted $\mathrm{do}(x_j)$, where $X_j$ is set to a fixed value $x_j$, leaving all other mechanisms unchanged [50]. The resulting distribution of any variable $X_i$ is known as the *interventional distribution* $p(X_i \mid \mathrm{do}(x_j))$. We use $\mathcal{D}_{\mathrm{obs}}$ to refer to datasets of independent and identically distributed (i.i.d) observations drawn from the model, and $\mathcal{D}_{\mathrm{int}}$ for i.i.d. observations drawn from any interventional distribution.

**Causal discovery and inference:** The task of causal discovery is to reconstruct the data generating graph $\mathcal{G}$ from an observational dataset $\mathcal{D}_{\mathrm{obs}}$ from an SCM. However, this typically only identifies a Markov equivalence class (MEC) of graphs that encode the same conditional independences [50]. Unique graph identification requires strong assumptions that may not hold in practice, such as hard restrictions on the allowable model classes [54, Ch. 4]. Furthermore, the identifiability guarantees of these methods also only hold in the infinite data setting. For a lot of tasks, causal discovery is a means to an end—namely, estimating interventional distributions for downstream tasks. With a causal graph and observational data, causal inference allows for estimating an interventional distribution $p(x_i|\mathrm{do}(x_j))$, if it is identifiable [50, 59][2]. However, inferring the ground truth causal graph is difficult. This has drastic implications for computing interventional distributions. Two graphs, even within the same MEC, may have very different interventional distributions [51]. Relying on a single causal graph to compute interventions can thus lead to incorrect conclusions.

**Bayesian causal inference:** Due to the limits of causal discovery, uncertainty is inherent in causal inference. Finite data issues further compound the problem. The Bayesian framework allows for quantifying the model uncertainty, both in the causal structure and functions, and use it for downstream decision making.

**Definition 2.1.** We define a *Bayesian causal model* (BCM) as the following hierarchical Bayesian model over causal graphs $\mathcal{G}$, functional mechanisms $\mathbf{f}$, and dataset $\mathcal{D}_{\mathrm{obs}}$ of $N_{\mathrm{obs}}$ samples: $\mathcal{G} \sim p_{\mathrm{BCM}}(\mathcal{G})$, $\mathbf{f} := \{f_i\}_{i \in V} \sim p_{\mathrm{BCM}}(\mathbf{f}|\mathcal{G})$, $\mathcal{D}_{\mathrm{obs}} := \{\mathbf{X}^n\}_{n=1}^{N_{\mathrm{obs}}} \sim \prod_{n=1}^{N_{\mathrm{obs}}} \prod_{i \in V} p_{\mathrm{BCM}}(x_i^n|f_i, \mathcal{G})$, where $x_i^n$ denotes the $i$-th node of the $n$-th observational sample. This implies the joint distribution $p_{\mathrm{BCM}}(\mathcal{D}_{\mathrm{obs}}, \mathbf{f}, \mathcal{G})$.

As BCMs are defined with a causal graph, they induce a distribution over interventional quantities as well. Analogous to standard causal models, interventions $p_{\mathrm{BCM}}(x_i|\mathrm{do}(x_j), f_i, \mathcal{G})$ can be computed by setting $f_j(\cdot) = x_j$ and leaving all other mechanisms unchanged.

Given an observational dataset, our task is to estimate an interventional distribution of interest. To do this, it is necessary to infer the possible graphs and functional mechanisms that generated the dataset. The Bayesian answer to this question is through the posterior

$$p_{\mathrm{BCM}}(\mathbf{f}, \mathcal{G}|\mathcal{D}_{\mathrm{obs}}) \propto p_{\mathrm{BCM}}(\mathcal{D}_{\mathrm{obs}}|\mathbf{f}, \mathcal{G}) p_{\mathrm{BCM}}(\mathbf{f}|\mathcal{G}) p_{\mathrm{BCM}}(\mathcal{G}). \qquad (2)$$

If the underlying model is identifiable, for example by restricting the function class of $\mathbf{f}$ [54, Ch. 4], then under suitable conditions[3] the posterior over $\mathcal{G}$ will concentrate on the true graph in the infinite data limit [13, 14, 12]. However, for finite data, or if the causal model is not identifiable, the posterior

---

[2]Note that we assume no hidden confounders which is common for causal discovery. Hence, given the ground truth causal structure, all interventional distributions are identifiable.

[3]The prior has to have positive density over the true underlying data generation process.

will quantify the uncertainty over causal graphs. To make use of the uncertainty, Bayes prescribes to average the interventions over the models [40], which we call the *posterior interventional distribution*

$$p_{\text{BCM}}(x_i|\text{do}(x_j), \mathcal{D}_{\text{obs}}) = \sum_{\mathcal{G}} \int p_{\text{BCM}}(x_i|\text{do}(x_j), \mathbf{f}, \mathcal{G}) p_{\text{BCM}}(\mathbf{f}|\mathcal{G}, \mathcal{D}_{\text{obs}}) p_{\text{BCM}}(\mathcal{G}|\mathcal{D}_{\text{obs}}) d\mathbf{f}. \quad (3)$$

Computing the above quantity is often intractable for two main reasons: 1) computing $p_{\text{BCM}}(\mathcal{G}|\mathcal{D}_{\text{obs}})$ is challenging as the number of causal graphs increases super-exponentially with the number of variables, 2) $p_{\text{BCM}}(\mathbf{f}|\mathcal{G}, \mathcal{D}_{\text{obs}})$ is only tractable for simple models.

**Transformer neural processes:** To bypass the need to compute the intermediate intractable quantities in eq. (3), we turn towards neural processes (NP)([20, 19]). From a Bayesian perspective, NPs incorporates a prior through the distribution over datasets that it is trained on [46], and, during inference, directly provides estimates of the posterior distribution of interest, side-stepping any explicit approximations of intermediate quantities. In particular, the Transformer Neural Process (TNP; [47, 16, 1]), which builds on the scalability and expressiveness of the transformer architecture [62], has achieved strong results across diverse domains [2, 63, 29, 14], motivating its use in our model-averaged approach for causal intervention estimation.

## 3 Related work

Estimating the posterior interventional distribution (eq. (3)) is challenging. The dominant paradigm involves a two stage process: 1) obtaining samples from the high dimensional posterior over graphs, and 2) estimating the interventional distribution under each sampled DAG, followed by averaging the result (fig. 1). Although principled, this process faces computational challenges in both stages.

The first stage is challenging due to the super-exponential size of the space of DAGs. Early score-based methods addressed this by leveraging score equivalence, allowing search over the small space of MECs instead of individual DAGs. To achieve this, they used restricted the model family to linear Gaussians, with specific priors to make the scores analytically tractable [26, 22]. To accommodate broader model classes, Madigan et al. [41] introduced an MCMC scheme over the space of DAGs. However, the large space of DAGs leads to slow mixing and convergence issues, limiting the number of effective posterior samples [17, 31, 58, 48, 33]. A common bottleneck in these approaches is that scoring the proposed structures at each MCMC step requires expensive marginal likelihood estimation. This is often mitigated through reducing the graph space by restricting the in-degree of each node. Variational inference (VI) offers a cheaper alternative [11] but struggles to capture multi-modal posteriors inherent in causal discovery [61, Sec. 3.1], and can still have a demanding computational costs (e.g. SVGD used in [38] scales quadratically with samples). Crucially, any inaccuracies or biases in this stage affects the downstream estimation in the second stage.

The second stage—averaging over the posterior of causal graphs—has its own significant computational burden. It requires performing inference with a potentially complex functional model for every single DAG sampled from the approximate posterior $p(\mathcal{G}|\mathcal{D}_{\text{obs}})$. As a result, previous work has only considered simple functional models where the inference is not too prohibitive. While early work in simple settings like linear Gaussian models allowed for closed-form averaging [64, 10], recent works often employ Gaussian Process (GP) networks [18] where this is not possible. To tackle this, Giudice et al. [24] use complex MCMC schemes for both hyperparameter posteriors of the GPs and graph sampling, but have to resort to approximating the final interventional posterior distribution with a Gaussian for computational tractability. Toth et al. [60, 61] also use GP networks but use the cheaper alternative of using MAP estimates for hyperparameters. However, both ultimately rely on the expensive process of estimating interventions by sampling from the GP posterior conditional on each DAG. Hence, despite variations, the core limitation of expensive inference persists across these approaches, especially prohibiting the use of more flexible function model classes.

In contrast to this explicit two-stage procedure, we propose leveraging NPs [20] to directly learn an estimator for the target interventional distribution conditional on the observational data $\mathcal{D}_{\text{obs}}$. Our approach aims to learn a mapping $\mathcal{D}_{\text{obs}} \mapsto p(y|\text{do}(x), \mathcal{D}_{\text{obs}})$ that does not require approximating potentially problematic intermediate quantities. This effectively amortises the complex inference and averaging procedure over the training of the NP. Our method thus seeks to mitigate the severe computational bottlenecks and avoid the compounding of approximation errors inherent in the standard two-stage pipeline for Bayesian causal inference.

There have been recent attempts at end-to-end or meta-learning approaches to estimating interventional distributions. While some methods assume knowledge of the ground truth causal graph [9, 42, 69], others offer a similar data driven approach as ours, but only in restricted settings. For example, Geffner et al. [21] offer an end-to-end approach but restrict to additive noise models, and do not perform functional inference. Sauter et al. [56] also use meta-learning to directly target the interventional distribution. Apart from differences in architecture and the loss used, their method is limited to discrete interventions. In contrast, we propose a general framework that is not restricted to types functional mechanisms or types of interventions. Further, by viewing meta-learning through a Bayesian lens, we provide insight into the role of the training data as encoding a prior distribution [25, 46, 28]. Tying our method to Bayesian inference also provides an understanding of the behaviour of our model under identifiability and non-identifiability of the causal model [12, 14, 13].

## 4 A transformer model for meta-learning causal inference

**Causal inference with neural processes:** The focus of this work is causal inference directly from data—predicting the distribution of a variable of interest $X_i$ when we intervene on another variable $\mathrm{do}(x_j)$ given access to only observational data $\mathcal{D}_{\mathrm{obs}}$ (eq. (3)). Instead of using a two-step approach as in fig. 1, which is computationally expensive and approximation error prone, we propose to *directly* learn the map from the observational dataset to the posterior interventional distribution of a BCM in an end-to-end fashion with NPs. With a chosen BCM, our aim is to approximate the true posterior interventional distribution defined in eq. 3. To do this, we minimise the expected Kullback-Leibler (KL) divergence over the tasks $\xi := (\mathcal{D}_{\mathrm{obs}}, i, j, X_j)$ between the true posterior interventional distribution and the NP model predictions $p_\theta(x_i|\mathrm{do}(x_j), \mathcal{D}_{\mathrm{obs}})$

$$\theta^* := \arg\min_\theta \mathbb{E}_\xi \left[ \mathrm{KL}(p_{\mathrm{BCM}}(X_i|\mathrm{do}(X_j), \mathcal{D}_{\mathrm{obs}}) \| p_\theta(X_i|\mathrm{do}(X_j), \mathcal{D}_{\mathrm{obs}})) \right]$$

$$= \arg\max_\theta \mathbb{E}_\xi[\mathbb{E}_{X_i|\xi}[\log(p_\theta(X_i|\mathrm{do}(X_j), \mathcal{D}_{\mathrm{obs}}))]] + C, \tag{4}$$

where $\xi \sim p(\mathcal{D}_{\mathrm{obs}}, i, j, x_j)$, $X_i|\xi \sim p_{\mathrm{BCM}}(x_i|\mathrm{do}(X_j), \mathcal{D}_{\mathrm{obs}})$, $\theta$ are the parameters of the NP, and $C$ is some constant independent of $\theta$.

Hence, our objective requires us to generate tasks, and interventional data from BCMs, in order to find the optimal $\theta^*$. To do this we, 1) sample a graph $\mathcal{G} \sim p_{\mathrm{BCM}}(\mathcal{G})$, and 2) a functional mechanism for each of the $D$ variables from the graph $\mathbf{f} \sim p_{\mathrm{BCM}}(\mathbf{f}|\mathcal{G})$. Conditioned on the sampled graph $\mathcal{G}$ and functional mechanism $\mathbf{f}$ we then 3) draw $N_{\mathrm{obs}}$ samples for each variable to construct the observational data $\mathcal{D}_{\mathrm{obs}} \sim P_{\mathrm{BCM}}(\mathcal{D}_{\mathrm{obs}}|\mathbf{f}, \mathcal{G})$. To construct the interventional data, keeping the same graph and functions as the observational data, we 4) randomly sample a variable index $j$ to intervene upon and $N_{\mathrm{int}}$ intervention values $\mathbf{X}_j \sim \mathcal{N}(\mathbf{0}, \mathbf{I})$, set the values of node $j$ to be $\mathbf{x}_j$, and 5) draw $N_{\mathrm{int}}$ samples of each node forming an interventional dataset $\mathcal{D}_{\mathrm{int}} \sim p_{\mathrm{BCM}}(\mathcal{D}_{\mathrm{int}}|\mathrm{do}(\mathbf{x}_j), \mathbf{f}, \mathcal{G})$. Finally we sample an outcome node index $i$ and extract samples of $p_{\mathrm{BCM}}(\mathbf{x}_i|\mathrm{do}(\mathbf{x}_j), f_i, \mathcal{G})$ from $\mathcal{D}_{\mathrm{int}}$.

While at training time we assume access to an explicitly specified Bayesian causal model, at test time we do not need access to such a model. In fact, the Bayesian prior is implicitly encoded into the NP through the distribution over its training datasets [46]. Inference for a new dataset simply requires a forward pass through the network.

**Recovery of exact prediction map:** Bruinsma [8, Proposition 3.26] shows that, in the limit of infinite tasks and model capacity, the global maximum of eq. (4) is achieved if and only if the model exactly learns the map $(\mathcal{D}_{\mathrm{obs}}, \mathbf{x}_j) \mapsto p_{\mathrm{BCM}}(x_i|\mathrm{do}(\mathbf{x}_j), \mathcal{D}_{\mathrm{obs}})$. Hence the NP learns to implicitly marginalises out any latent variables in eq. (3) [25]. While the constraint of infinite tasks is limiting when applying NP to real-world datasets, if the tasks are generated through a known Bayesian causal model, we have in theory access to an infinite amount of tasks.

**Model architecture and desirable properties:** Given we are interested in predicting $p_{\mathrm{BCM}}(x_i|\mathrm{do}(\mathbf{x}_j), \mathcal{D}_{\mathrm{obs}})$, variables play distinct roles as either the outcome node $X_i$, the intervening node $X_j$, or the nodes that are being marginalised. Thus, properties of this distribution, and the role of the variables, guide our architecture choice. First, the interventional distribution remains invariant when the observational data samples are permuted or the nodes being marginalised over are permuted (*permutation-invariance* with respect to observational samples and to all nodes except the outcome $X_i$ and intervention $X_j$). Second, permuting the interventional queries should permute the samples of the target distribution accordingly (*permutation-equivariance* with respect

to interventional samples). Similarly, permuting any nodes involving the outcome or intervention nodes should yield the corresponding permuted interventional distribution (*permutation-equivariance with respect to outcome and intervention nodes*). For example, permutting the outcome and intervention nodes $i \leftrightarrow j$ should result in the permuted $p(\mathbf{x}_j|\mathrm{do}(\mathbf{x}_i), \mathcal{D}_{\mathrm{obs}})$. Furthermore, we assume no correlations among the interventional samples and as such restrict our attention to the family of conditional neural processes (CNPs), where the predictive distributions factorises over the interventional samples $p_\theta(\mathbf{x}_i|\mathrm{do}(\mathbf{x}_j), \mathcal{D}_{\mathrm{obs}}) = \prod_{n=1}^{N_{\mathrm{int}}} p_\theta(x_i^n|\mathrm{do}(x_j^n), \mathcal{D}_{\mathrm{obs}})$. As interventional distributions can be non-Gaussian even in very simple cases, we opt for a Mixture of Gaussians (MoG) representation of $p_\theta(x_i|\mathrm{do}(x_j), \mathcal{D}_{\mathrm{obs}})$ [6].

An architecture that is flexible enough to satisfy these desiderata is the transformer [62, 34]. We provide a schematic architecture for our proposed model, the *Model-Averaged Causal Estimation Transformer Neural Process* (MACE-TNP), in fig. 2, and give a detailed explanation of each of its components in Appendix A.2.

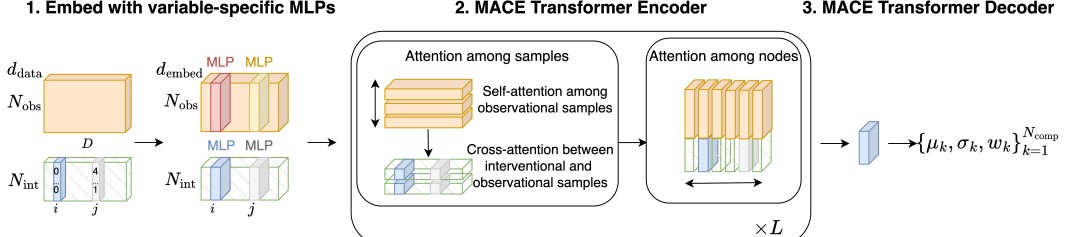

Figure 2: Overview of MACE-TNP yielding $p_\theta(\mathbf{x}_i|\mathrm{do}(\mathbf{x}_j), \mathcal{D}_{\mathrm{obs}})$. Inputs are 1) embedded via variable-specific MLPs, 2) fed into a transformer encoder that alternates sample-wise and node-wise attention. The resulting outcome node representation from the unknown interventional distribution is 3) decoded to obtain the parameters of the NP distribution.

**Embedding:** The model takes as input a matrix of $N_{\mathrm{obs}}$ observational samples of $D$ nodes and an intervention matrix of $N_{\mathrm{int}}$ queries for a node of interest $X_j$, with the rest of the $D-1$ nodes masked out (by zeroing them out). Variables play distinct roles in both matrices, requiring different encoding strategies—either as the node we intervene upon ($j$), outcome node ($i$), or node we marginalise over. Our input representation also needs to reflect that observational and interventional samples originate from different distributions. As such, we employ six variable-specific MLPs, one for each combination of (node type, sample type). These MLPs produce $d_{\mathrm{embed}}$-dimensional embeddings, resulting in representations $\mathbf{Z}_{\mathrm{obs}} \in \mathbb{R}^{N_{\mathrm{obs}} \times D \times d_{\mathrm{embed}}}$ and $\mathbf{Z}_{\mathrm{int}} \in \mathbb{R}^{N_{\mathrm{int}} \times D \times d_{\mathrm{embed}}}$ for the observational and interventional data, respectively.

**MACE transformer encoder:** To satisfy the afore-mentioned permutation symmetries, we construct an encoder of $L$ layers where we alternate between attention among samples, followed by attention among nodes [15, 39, 32]. The two attention mechanisms that we use are multi-head self-attention (MHSA) and multi-head cross-attention (MHCA)—both defined in eq. (5) and eq. (7), respectively. More specifically, at each layer $l \in \{1, \ldots, L\}$ we 1) update the observational data representation $\mathbf{Z}_{\mathrm{obs}}^l \in \mathbb{R}^{N_{\mathrm{obs}} \times D \times d_{\mathrm{embed}}}$ through MHSA. This is then used to 2) modulate the interventional data representation $\mathbf{Z}_{\mathrm{int}}^l \in \mathbb{R}^{N_{\mathrm{int}} \times D \times d_{\mathrm{embed}}}$ through MHCA, an operation that assures permutation equivariance with respect to the interventional samples. We then 3) concatenate the two representations to obtain $\mathbf{Z}^{l'} \in \mathbb{R}^{(N_{\mathrm{obs}}+N_{\mathrm{int}}) \times D \times d_{\mathrm{embed}}}$, followed by 4) MHSA among the nodes to yield the output at layer $l$ which acts as input at layer $l+1$:

$$\underbrace{1.\mathbf{Z}_{\mathrm{obs}}^l = \mathrm{MHSA}(\mathbf{Z}_{\mathrm{obs}}^l) \to 2.\mathbf{Z}_{\mathrm{int}}^l = \mathrm{MHCA}(\mathbf{Z}_{\mathrm{int}}^l, \mathbf{Z}_{\mathrm{obs}}^l)}_{\text{attention among samples}} \to 3.\mathbf{Z}^{l'} = [\mathbf{Z}_{\mathrm{obs}}^l, \mathbf{Z}_{\mathrm{int}}^l] \to \underbrace{4.[\mathbf{Z}_{\mathrm{obs}}^{l+1}, \mathbf{Z}_{\mathrm{int}}^{l+1}] = \mathrm{MHSA}(\mathbf{Z}^{l'})}_{\text{attention among nodes}}.$$

**MACE decoder:** The information required for the target distribution is now encoded in the outcome node ($i$-th index) of the interventional matrix, $\mathbf{Z}_{\mathrm{int},i}^L \in \mathbb{R}^{N_{\mathrm{int}} \times d_{\mathrm{embed}}}$. This is passed through an MLP decoder to obtain the final distribution. To parametrise expressive interventional distributions, we construct the output distribution of the NP as an MoG with $N_{\mathrm{comp}}$ components [6]. The NP outputs the mean, standard deviation and weight corresponding to each component for each interventional query $x_j^n$: $\{\boldsymbol{\mu}, \boldsymbol{\sigma}, \mathbf{w}\}(x_j^n) := \{\mu_k(x_j^n), \sigma_k(x_j^n), w_k(x_j^n)\}_{k=1}^{N_{\mathrm{comp}}}$.

**Loss:** The model is trained to maximise the log-posterior interventional distribution according to eq. (4), where, with a MoG parameterisation:

$$\mathcal{L}_\theta(\mathbf{x}_i, \{\boldsymbol{\mu}, \boldsymbol{\sigma}, \mathbf{w}\}(\mathbf{x}_j)) = \sum_{n=1}^{N_{\text{int}}} \log p_\theta(x_i^n \,|\, \text{do}(x_j^n), \mathcal{D}_{\text{obs}}) = \sum_{n=1}^{N_{\text{int}}} \log \left( \sum_{k=1}^{N_{\text{comp}}} w_k(x_j^n) \cdot \mathcal{N}(x_i^n \,|\, \mu_k(x_j^n), \sigma_k^2(x_j^n)) \right)$$

## 5   Experiments

We evaluate the performance of our model, MACE-TNP, against Bayesian causal inference baselines, and a causal discovery method that selects a single graph. With our experiments we aim to answer: 1) When analytically tractable, can we confirm that our model recovers the true posterior interventional distribution under identifiability and non-identifiability of the causal graph, 2) How does our model compare against baselines when the baselines' assumptions are *respected* and when they are *violated*, 3) How does our model perform when the number of nodes are scaled, 4) How does our model perform when we do not have knowledge of the data generating process? Code for our experiments is available at: `https://github.com/Anish144/CausalInferenceNeuralProcess`.

To train MACE-TNP, we randomise the number of observational samples $N_{\text{obs}} \sim \mathcal{U}\{50, 750\}$, and set $N_{\text{int}} = 1000 - N_{\text{obs}}$. The training loss is evaluated on these $N_{\text{int}}$ samples. For testing, we sample 500 observation points and compute the loss against 500 intervention points.

**Baselines:** We benchmark against methods that infer distributions over causal graphs and sample to marginalise across these graphs when estimating posterior interventional distributions. DiBS-GP [60], ARCO-GP [61], and BCI-GPN [24] all use GP networks, but differ in the inference procedure over graphs. DiBS-GP uses a continuous latent to parametrise a graph, ARCO-GP uses an order parametrisation of DAGs, whereas BCI-GPN uses an MCMC scheme to sample DAGs. We also compare against DECI [21], which assumes additive noise and uses autoregressive neural networks to learn a distribution over causal graphs while only learning point estimates for functions. Finally, to show that learning a distribution over graphs is useful for causal inference, we compare against a non-Bayesian baseline that uses NOGAM [44] to infer a single DAG, and estimates the interventional distribution by using GPs: NOGAM-GP.

**Metrics:** For evaluation, we compare the model's posterior interventional distributions on held out datasets. Unlike graph-based metrics that only assess structural accuracy, this task requires correct inference of both the causal graph as well as the functional mechanisms. The posterior interventional distribution of the data generating model is only analytically tractable in simple cases (section 5.1). In these instances, we report the KL divergence between the data generating model's posterior interventional distribution and that of the NP, averaged over intervention queries. When the analytical solution is not available, we report the negative log-posterior interventional density (NLPID) of the true intervention outcomes under the model: $-\mathbb{E}_{X_j \sim \mathcal{N}(0,1)}[\mathbb{E}_{p_{\text{BCM}}(x_i|\text{do}(x_j), \mathcal{D})}[\log p_\theta(X_i|\text{do}(X_j), \mathcal{D})]]$ [23].

### 5.1   Two-node linear Gaussian model

First, we test on synthetic data where the ground-truth posterior interventional distribution is tractable. Specifically, we are interested in the behaviour of our model under identifiability and non-identifiability of the causal structure. For this, we generate from a bivariate, single edge, linear model, with Gaussian noise—a model that is identifiable when the variances are known [53], but non-identifiable under specific priors [22] (appendix B.1). Besides the KL between the ground-truth and MACE-TNP, we also report the KL between the interventional distribution conditioned on the true function and graph, and MACE-TNP's output distribution. The latter gauges accuracy in learning the true interventional distribution, which requires identifiability of the causal graph.

The results are shown in fig. 3 for the identifiable (left) and non-identifiable (right) cases. They confirm that the output of MACE-TNP does indeed converge to the Bayesian optimal posterior, as the dark blue lines indicating $\text{KL}(p_{BCM}(x_i|\text{do}(x_j), \mathcal{D}_{\text{obs}})\|p_\theta(x_i|\text{do}(x_j), \mathcal{D}_{\text{obs}}))$ go to 0 with increasing sample size in both cases. Moreover, as expected, $\text{KL}(p_{BCM}(x_i|\text{do}(x_j), \mathbf{f}^*, \mathcal{G}^*)\|p_\theta(x_i|\text{do}(x_j), \mathcal{D}_{\text{obs}}))$, where $\{\mathbf{f}^*, \mathcal{G}^*\}$ characterise the true data-generating mechanism, does not go to 0 in the second case due to the non-identifiability of the causal graph (as indicated by the red line). The flexibility of our architecture also allows for conditional queries, multiple interventions, as well as easily incorporating

interventional data to help identify causal relations. Hence, we investigate here whether providing a small number $M_{\text{int}} = 5$ of true interventional samples, alongside the observational data, resolves identifiability challenges in the non-identifiablee case. As shown in fig. 3 (right) with the green line, we find that this does indeed lower the $\text{KL}(p_{BCM}(x_i|\text{do}(x_j), \mathbf{f}^*, \mathcal{G}^*)\|p_\theta(x_i|\text{do}(x_j), \mathcal{D}_{\text{obs}}, \{x_i^n\}_{n=1}^{M_{\text{int}}}))$, suggesting that even limited interventions can enhance identifiability. Moreover, we show that the divergence further decreases with more interventional samples $M_{\text{int}}$ in appendix B.1.

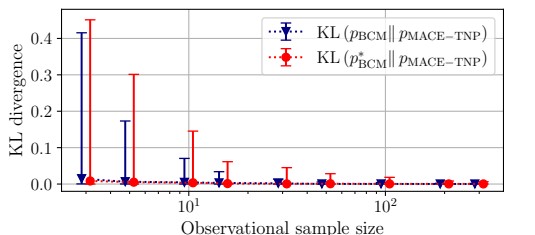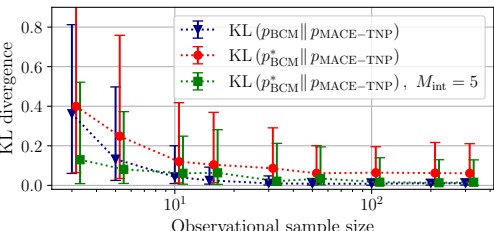

Figure 3: KL divergences as a function of the observational sample size, for the identifiable case (left) and the non-identifiable one (right). Dark blue denotes $p_{\text{BCM}}$—the posterior interventional distribution defined in eq. (3), red and green use $p_{BCM}^*$—the interventional distribution conditioned on $\{\mathbf{f}^*, \mathcal{G}^*\}$. We additionally provide MACE-TNP with $M_{\text{int}} = 5$ interventional samples. We indicate the median and the 10-90% quantiles.

## 5.2 Three-node linear Gaussian confounder vs. mediator

In the previous section, we showed that in the two-node case, when the causal graph is identifiable, MACE-TNP can identify the correct interventional distribution. However, with more variables, finding the correct interventional distribution requires proper adjustment of the variables that are not the treatment or the outcome. For example, if we are interested in the distribution $p(y|\text{do}(x))$ and another variable $z$ is a confounder, the interventional distribution is $p(y|\text{do}(x)) = \int p(y|x, z)p(z)\mathrm{d}z$. However, if the variable $z$ is a mediator, the interventional distribution is $p(y|\text{do}(x)) = \int p(y|x, z)p(z|x)\mathrm{d}z$. Here, we show that in the three-node case, with identifiable causal structures, MACE-TNP *implicitly* performs the required adjustment.

We train a MACE-TNP model on fully-connected three-node graphs with data generated from identifiable linear Gaussian structures [53]. We then estimate the KL divergence between the interventional distribution conditioned on the true data-generating mechanism and the model's interventional distribution: $\text{KL}(p_{BCM}(x_i|\text{do}(x_j), \mathbf{f}^*, \mathcal{G}^*)\|p_\theta(x_i|\text{do}(x_j), \mathcal{D}_{\text{obs}}))$ for 1) confounder graphs, 2) mediator graphs, and 3) confounder graphs where the confounder is unobserved in the context. Figure 4 (right) shows the results. As expected, as we increase the sample size for 1) and 2), MACE-TNP more accurately identifies the correct interventional distribution, showing that it is implicitly adjusting the third variable depending on whether it is a mediator or a confounder. When the confounder is unobserved, and the interventional distribution and the causal graph are not identifiable, the KL between the true distribution and the MACE-TNP tends to a constant value above zero.

## 5.3 Three-node experiments

Next, we compare our model MACE-TNP against the baselines in a three-node setting. Here, there are 25 graphs in total, making inference over the graph easier than in higher-node settings. Given that most baselines either use neural networks or GPs, we compare MACE-TNP to the baselines under two scenarios: 1) when tested on GP data, and 2) when tested on data generated using neural networks (NN). For each functional mechanism, we train a separate MACE-TNP model. Full experimental details are provided in Appendix C.2.2.

When tested in-distribution (on datasets from the same distribution the model was trained on), MACE-TNP consistently outperforms all baselines across both functional mechanisms, as shown in table 1. MACE-TNP outperforms GP-based methods through its implicit handling of hyperparameter inference, that the GP baselines may struggle with. It also surpasses DECI (an NN-based approach) on both GP and NN data by employing a Bayesian treatment over functions. This highlights a key

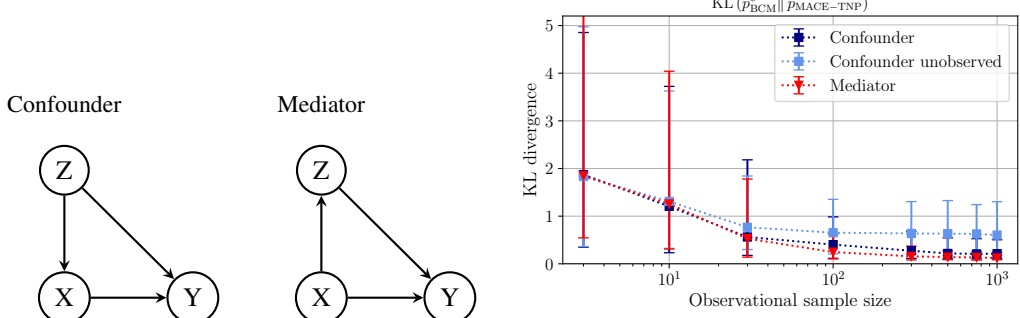

Figure 4: KL divergence (right) of the interventional distribution conditioned on $\{\mathbf{f}^*, \mathcal{G}^*\}$ and the model's for the confounder (dark blue), mediator (red) and unobserved confounder (light blue) cases. With increasing sample size, MACE-TNP identifies the correct distribution for both the mediator and confounder cases, implicitly carrying out the required adjustment.

advantage of MACE-TNP: it can easily incorporate complex Bayesian causal models into its training pipeline by sampling training datasets, whereas traditional Bayesian methods rely on inadequate approximations for complex scenarios.

Table 1: Results for MACE-TNP and baselines on the three-node experiments. We show the NLPID ($\downarrow$) and report the mean $\pm$ the error of the mean over 100 datasets. Each row corresponds to a different functional mechanism used in the test set (GP / NN).

|    | MACE-TNP | DiBS-GP | ARCO-GP | BCI-GPN | DECI | NOGAM+GP |
|----|----------|---------|---------|---------|------|----------|
| GP | $\mathbf{563.9 \pm 23.4}$ | $644.2 \pm 27.2$ | $630.7 \pm 22.3$ | $628.5 \pm 27.5$ | $632.0 \pm 25.6$ | $749.4 \pm 43.0$ |
| NN | $\mathbf{527.9 \pm 19.8}$ | $807.6 \pm 50.1$ | $851.2 \pm 55.0$ | $706.8 \pm 5.0$ | $588.0 \pm 23.6$ | $815.6 \pm 58.4$ |

**Out-of-distribution testing:** All previous experiments trained MACE-TNP on the true functional mechanism (GP or NN). However, a natural question is: how does the model perform when tested on out-of-distribution (OOD) data? To probe this, we evaluate MACE-TNP (GP) on NN-generated data and MACE-TNP (NN) on GP-generated data. As expected, performance degrades when the test mechanism differs from training, since our model lacks built-in inductive bias for unseen mechanisms: MACE-TNP (GP) tested on NN achieves $608.3 \pm 17.3$, compared to MACE-TNP (NN) with $527.9 \pm 19.8$. Similarly, MACE-TNP (NN) tested on GP achieves $678.0 \pm 10.0$, compared to MACE-TNP (GP) with $563.9 \pm 23.4$. However, NPs trivially support additional training on any data likely to be informative. Indeed, training MACE-TNP on the combined GP+NN data nearly recovers in-distribution accuracy, achieving $531.0 \pm 19.4$ on NN data and $583.9 \pm 21.5$ on GP data (see table 4).

## 5.4 Higher dimensional experiment

We next investigate the scalability of our method. This is especially relevant in many modern applications (e.g. genomics, neuroscience, econometrics, and social-network analysis) which naturally involve high-dimensional systems where understanding intervention effects is crucial. We do so by testing a single trained model on increasingly higher-dimensional data, scaling up from 20 up to 40 nodes. The functional mechanisms used are a mix of NNs and functions drawn from a GP with an additional latent variable input. More details on data generation are given in appendix B.3.

Table 2 shows that MACE-TNP outperforms both the Bayesian, as well as the non-Bayesian baselines across all node sizes. Moreover, the underperformance of the non-Bayesian baseline NOGAM+GP underscores the importance of capturing uncertainty with a higher number of variables. The majority of our baselines involve GP-based approaches, which can become prohibitively expensive with a higher number of variables. For example, we do not report BCI-GPN as its MCMC scheme is too expensive for these node sizes. In contrast, MACE-TNP can readily leverage advancements that have made neural network architectures scale favourably in other domains. Inference after training only requires a forward pass through the network.

Table 2: Results for MACE-TNP and baselines on the higher dimension experiment. We show the NLPID ($\downarrow$) and report the mean $\pm$ the error of the mean over 100 datasets. Each row corresponds to a different number of variables.

|  | MACE-TNP | DiBS-GP | ARCO-GP | DECI | NOGAM+GP |
|---|---|---|---|---|---|
| 20 variables | **660.4 $\pm$ 5.2** | 701.9 $\pm$ 4.0 | 701.9 $\pm$ 4.0 | 686.3 $\pm$ 6.7 | 942.7 $\pm$ 23.8 |
| 30 variables | **653.3 $\pm$ 5.7** | 713.2 $\pm$ 4.7 | 713.0 $\pm$ 4.7 | 675.6 $\pm$ 6.1 | 946.9 $\pm$ 19.2 |
| 40 variables | **665.8 $\pm$ 4.8** | 711.5 $\pm$ 4.6 | 712.1 $\pm$ 4.6 | 683.0 $\pm$ 5.1 | 986.0 $\pm$ 20.0 |

## 5.5 Unknown dataset generation process

Finally, we apply our proposed method on the Sachs proteomics dataset [55], which includes measurements of $D = 11$ proteins from thousands of cells under various molecular interventions. Crucially, we do not retrain any model for this task; instead, we reuse the model from section 5.4 which was trained exclusively on synthetic data. Following [7, 36, 66], we retain only samples with interventions directly targeting one of the $D = 11$ proteins, yielding 5846 samples: 1755 observational and 4091 interventional across five single-protein perturbations. The results indicate that our method performs competitively with our strongest baseline, giving an NLPID for MACE-TNP of $998.9 \pm 104.9$ compared to $1000.9 \pm 133.5$ for DECI, when averaged over 5 interventional queries and 10 outcome nodes. We provide additional comparisons to the remaining baselines in appendix C.2.3. This shows the potential of tackling interventional queries in real-world settings with a fast, data-driven framework that captures uncertainty in a principled manner and leverages flexible and expressive neural network architectures.

## 6 Conclusions

We address the challenge of efficiently estimating interventional distributions when the causal graph structure is unknown. Our solution, MACE-TNP, is an end-to-end meta-learning framework that directly approximates the Bayesian model-averaged interventional distribution by mapping observational data to posterior interventional distributions. When the true posterior is available, we show empirically that the model's predictions converge to it. Moreover, in a simple non-identifiable case, we show that interventional data allows for capturing the true underlying mechanism. When employing more complex functional mechanisms, as well as higher-dimensional data (up to 40 nodes), MACE-TNP outperforms strong Bayesian and non-Bayesian baselines, with the only requirement being access to samples from a prior distribution (implicit or explicit) at meta-train time. One limitation of our model is its reliance on substantial training-time compute and data to effectively capture a diverse range of causal mechanisms. Moreover, if the test distribution is not properly covered by the training data distribution, the model may struggle to generalise. However, as our out-of-distribution experiments demonstrate, integrating additional data to better cover the target distribution is straightforward and efficient in improving MACE-TNP's generalisation capabilities. Finally, the attention mechanism scales quadratically with the number of variables and samples, which can be costly. However, MACE-TNP can leverage recent advances in sparse and low-rank attention to mitigate this overhead [67, 68]. Future work includes a thorough investigation into the interventional sample complexity required by the NP for accurate interventional estimation, as well as how to best construct the prior over BCMs to capture the complexities of real-life data.

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

# A Architecture

This section provides the definitions of the architectures described in the main paper.

Transformers [62] can be viewed as general set functions [34], making them ideally suited for NPs, which must ingest datasets. We begin by briefly overviewing transformers, defining the attention operations and how we construct a transformer layer, followed by how we integrate transformers into the MACE-TNP architecture.

## A.1 Transformers

**MHSA and MHCA** Throughout this work we make use of two operations: multi-head self-attention (MHSA) and multi-head cross-attention (MHCA). Let $\mathbf{Z} \in \mathbb{R}^{N \times D_z}$ be a set of $N$ tokens of dimensionality $D_z$. Then, for $\forall\, n = 1, \ldots, N$, the MHSA operation updates this set of tokens as follows

$$\mathbf{z}_n \leftarrow \mathrm{cat}\Big(\Big\{ \sum_{m=1}^{N} \alpha_h(\mathbf{z}_n, \mathbf{z}_m)\mathbf{z}_m^T \mathbf{W}_{V,h} \Big\}_{h=1}^{H}\Big)\mathbf{W}_O, \tag{5}$$

where $\mathbf{W}_{V,h} \in \mathbb{R}^{D_z \times D_V}$ and $\mathbf{W}_O \in \mathbb{R}^{HD_V \times D_z}$ are the value and projection weight matrices, $H$ denotes the number of heads, and $\alpha_h$ is the attention mechanism. We opt for the most widely used softmax formulation

$$\alpha_h(\mathbf{z}_n, \mathbf{z}_m) = \mathrm{softmax}(\{\mathbf{z}_n^T \mathbf{W}_{Q,h} \mathbf{W}_{K,h}^T \mathbf{z}_m\}_{m=1}^{N})_m, \tag{6}$$

where $\mathbf{W}_{Q,h} \in \mathbb{R}^{D_z \times D_{QK}}$ and $\mathbf{W}_{K,h} \in \mathbb{R}^{D_z \times D_{QK}}$ are the query and key matrices.

The MHCA operation performs attention between two *different* sets of tokens $\mathbf{Z}_1 \in \mathbb{R}^{N_1 \times D_z}$ and $\mathbf{Z}_2 \in \mathbb{R}^{N_2 \times D_z}$. For $\forall\, n = 1, \ldots, N_1$, the following update on $\mathbf{z}_{1,n}$ is performed:

$$\mathbf{z}_{1,n} \leftarrow \mathrm{cat}\Big(\Big\{ \sum_{m=1}^{N_2} \alpha_h(\mathbf{z}_{1,n}, \mathbf{z}_{2,m})\mathbf{z}_{2,m}^T \mathbf{W}_{V,h} \Big\}_{h=1}^{H}\Big)\mathbf{W}_O. \tag{7}$$

In order to obtain the attention blocks used within the transformer, these operations are typically combined with residual connections, layer-isations and point-wise MLPs.

More specifically, we define the MHSA operation as follows:

$$\begin{aligned}
\widetilde{\mathbf{Z}} &\leftarrow \mathbf{Z} + \mathrm{MHSA}(\text{layer-norm}_1(\mathbf{Z})) \\
\mathbf{Z} &\leftarrow \widetilde{\mathbf{Z}} + \mathrm{MLP}(\text{layer-norm}_2(\widetilde{\mathbf{Z}})).
\end{aligned} \tag{8}$$

Similarly, the MHCA operation is defined as:

$$\begin{aligned}
\widetilde{\mathbf{Z}}_1 &\leftarrow \mathbf{Z}_1 + \mathrm{MHCA}(\text{layer-norm}_1(\mathbf{Z}_1), \text{layer-norm}_1(\mathbf{Z}_2)) \\
\mathbf{Z}_1 &\leftarrow \widetilde{\mathbf{Z}}_1 + \mathrm{MLP}(\text{layer-norm}_2(\widetilde{\mathbf{Z}}_1)).
\end{aligned} \tag{9}$$

**Masked-MHSA** Consider the general case in which we want to update $N$ token $\mathbf{Z} \in \mathbb{R}^{N \times D_z}$. There might be some situations where we want to make the update of a certain token $\mathbf{z}_n \in \mathbf{Z}$ independent of some other tokens. In that case, we can specify a set $M_n \subseteq \mathbb{N}_{\leq N}^{+}$ containing the indices of the tokens we want to make the update of $\mathbf{z}_n$ independent of. Then, we can modify the pre-softmax activations within the attention mechanism $\tilde{\alpha}_h(\mathbf{z}_n, \mathbf{z}_m)$, where $\alpha_h(\mathbf{z}_n, \mathbf{z}_m) = \mathrm{softmax}(\tilde{\alpha}_h(\mathbf{z}_n, \mathbf{z}_m))$ as follows:

$$\tilde{\alpha}_h(\mathbf{z}_n, \mathbf{z}_m) = \begin{cases} -\infty & \text{if } m \in M_n \\ \mathbf{z}_n^T \mathbf{W}_{Q,h} \mathbf{W}_{K,h}^T \mathbf{z}_m & \text{otherwise} \end{cases} \tag{10}$$

From the indices of $M_n$ we can construct a binary masking matrix $\mathbf{M} \in \{0,1\}^{N \times N}$:

$$\mathbf{M}_{n,m} = \begin{cases} 0 & \text{if } m \in M_n \\ 1 & \text{otherwise} \end{cases}$$

When used in the context of MHSA, we refer to this operation as masked-MHSA and represent it as $\mathbf{Z} = \mathrm{masked\text{-}MHSA}(\mathbf{Z}, \mathbf{M})$.

## A.2 Model-Averaged Causal Estimation Transformer Neural Processes (MACE-TNPs)

We refer to Nguyen and Grover [47], Ashman et al. [2] for a complete description of standard TNP architectures, and focus on describing the architecture of the MACE-TNP in more detail. Our proposed architecture is conceptually similar to the standard TNP architectures, but incorporates specific design choices and inductive biases that make it suitable for causal estimation.

We assume we have access to $N_{\text{obs}}$ observational samples and want to predict the distribution of $N_{\text{int}}$ interventional samples. The inputs to the MACE-TNP are: the observational dataset $\mathcal{D}_{\text{obs}} \in \mathbb{R}^{N_{\text{obs}} \times D \times d_{\text{data}}}$, the values of the node we intervene upon $\mathbf{x}_j \in \mathbb{R}^{N_{\text{int}}}$ (implying we intervene on node $j$), and the outcome node index $i$. Let $\mathcal{D}_{\text{obs},i} \in \mathbb{R}^{N_{\text{obs}} \times d_{\text{data}}}$ denote the observational data at node $i$. We omit the batch dimension for notational convenience.

**Data pre-processing** The model takes as input a matrix of $N_{\text{obs}}$ observational samples of $D$ nodes and an intervention matrix of $N_{\text{int}}$ queries for a node of interest $X_j$, with the rest of the $D - 1$ nodes masked out (by zeroing them out). Let $\mathcal{D}_{\text{int},i} \in \mathbb{R}^{N_{\text{int}} \times d_{\text{data}}}$ denote the interventional data at node $i$. In the following we use $\mathcal{D}_{\text{obs},\{k \in [D] \setminus \{i,j\}\}}$ to denote nodes in the observational dataset that are being marginalised over.

**Embedding** To differentiate between the different type of variables, we employ six different types of encodings, depending on the source of the data (observational (obs) or interventional (int)), and the type of the node (node we intervene upon ($j$), outcome node ($i$), or node we marginalise over). These are all performed using 2-layer MLPs of dimension $d_{\text{embed}}$.

$$
\begin{aligned}
\text{observational, intervention node:} \quad & \mathbf{Z}_{\text{obs},j} = \text{MLP}_{\text{obs},j}(\mathcal{D}_{\text{obs},j}) && (11)\\
\text{observational, outcome node:} \quad & \mathbf{Z}_{\text{obs},i} = \text{MLP}_{\text{obs},i}(\mathcal{D}_{\text{obs}}, i)\\
\text{observational, marginal nodes:} \quad & \mathbf{Z}_{\text{obs},\{k \in [D] \setminus \{i,j\}\}} = \text{MLP}_{\text{obs}}(\mathcal{D}_{\text{obs},\{k \in [D] \setminus \{i,j\}\}})\\
\text{interventional, intervention node:} \quad & \mathbf{Z}_{\text{int},j} = \text{MLP}_{\text{int},j}(\mathcal{D}_{\text{int},j})\\
\text{interventional, outcome node:} \quad & \mathbf{Z}_{\text{int},i} = \text{MLP}_{\text{int},i}(\mathcal{D}_{\text{int},i})\\
\text{interventional, marginal nodes:} \quad & \mathbf{Z}_{\text{int},\{k \in [D] \setminus \{i,j\}\}} = \text{MLP}_{\text{int}}(\mathcal{D}_{\text{int},\{k \in [D] \setminus \{i,j\}\}}),
\end{aligned}
$$

where $\{k \in [D] \setminus \{i,j\}\}$ represents the set of indices from $\{1, \ldots, D\}$ excluding $i$ and $j$. The representations are then concatenated back together in the original node order:

$$
\mathbf{Z}_{\text{obs}} = \text{concat}\left([\mathbf{Z}_{\text{obs},k}]_{k \in [D]}\right), \quad \text{where } \mathcal{D}_k = \begin{cases} \mathbf{Z}_{\text{obs},i} & \text{if } k = i \\ \mathbf{Z}_{\text{obs},j} & \text{if } k = j \\ \mathbf{Z}_{\text{obs},k} & \text{otherwise} \end{cases}
$$

$$
\mathbf{Z}_{\text{int}} = \text{concat}\left([\mathbf{Z}_{\text{int},k}]_{k \in [D]}\right), \quad \text{where } \mathbf{Z}_k = \begin{cases} \mathbf{Z}_{\text{int},i} & \text{if } k = i \\ \mathbf{Z}_{\text{int},j} & \text{if } k = j \\ \mathbf{Z}_{\text{int},k} & \text{otherwise} \end{cases}
$$

After the embedding stage, we obtain the representation of the observational dataset $\mathbf{Z}_{\text{obs}} \in \mathbb{R}^{N_{\text{obs}} \times D \times d_{\text{embed}}}$, and the representation of the interventional one $\mathbf{Z}_{\text{int}} \in \mathbb{R}^{N_{\text{int}} \times D \times d_{\text{embed}}}$.

**MACE Transformer Encoder** We utilise a transformer-based architecture composed of $L$ layers, where we alternate between attention among samples, followed by attention among nodes. This choice preserves 1) permutation-invariance with respect to the obervational samples, 2) permutation-equivariance with respect to the interventional samples, 3) permutation-invariance with respect to the nodes we marginalise over, and 4) permutation-equivariance with respect to the outcome and interventional nodes. Although we generally omit the batch dimension for convenience, we include it in this subsection to accurately reflect our implementation. Thus, the input to the MACE transformer encoder are the observational data representation $\mathbf{Z}_{\text{obs}} \in \mathbb{R}^{B \times N_{\text{obs}} \times D \times d_{\text{embed}}}$ and interventional data representation $\mathbf{Z}_{\text{int}} \in \mathbb{R}^{B \times N_{\text{int}} \times D \times d_{\text{embed}}}$, with $B$ the batch size.

**Attention among samples** We propose two variants to perform attention among samples. We use the less costly MHSA + MHCA variant for the experiments in the main paper and show that it performs better in appendix C.2.2.

1. **Masked-MHSA** among the observational and interventional samples: At each layer $l$, we first move the node dimension to the batch dimension for efficient batched attention: $\mathbf{Z}_{\text{obs}}^l \in \mathbb{R}^{B \times N_{\text{obs}} \times D \times d_{\text{embed}}} \rightarrow \mathbb{R}^{(B \times D) \times N_{\text{obs}} \times d_{\text{embed}}}$ and $\mathbf{Z}_{\text{int}}^l \in \mathbb{R}^{B \times N_{\text{int}} \times D \times d_{\text{embed}}} \rightarrow \mathbb{R}^{(B \times D) \times N_{\text{int}} \times d_{\text{embed}}}$. We then concatenate the two representations $\mathbf{Z}^l \in \mathbb{R}^{(B \times D) \times (N_{\text{obs}} + N_{\text{int}}) \times d_{\text{embed}}} = [\mathbf{Z}_{\text{obs}}^l, \mathbf{Z}_{\text{int}}^l]$, and construct a mask $\mathbf{M} \in \mathbb{R}^{N_{\text{obs}} + N_{\text{int}}}$ that only allows interventional tokens to attend to observational ones.

$$\mathbf{M}_{n,m} = \begin{cases} 1 & \text{if } m < N_{\text{obs}} \\ 0 & \text{otherwise} \end{cases}$$

   We then perform masked-MHSA: $\mathbf{Z}^l = \text{masked-MHSA}(\mathbf{Z}^l, \mathbf{M})$. This strategy has a computational complexity $\mathcal{O}((N_{\text{obs}} + N_{\text{int}})^2)$.

2. **MHSA + MHCA**: An alternative, less costly strategy, is to perform MHSA on the observational data, followed by MHCA between the interventional and observational data. More specifically, as in the previous case we move the node dimension to the batch dimension and then perform:

$$\mathbf{Z}_{\text{obs}}^l = \text{MHSA}(\mathbf{Z}_{\text{obs}}^l)$$
$$\mathbf{Z}_{\text{int}}^l = \text{MHCA}(\mathbf{Z}_{\text{int}}^l, \mathbf{Z}_{\text{obs}}^l).$$

   We then concatenate the two representations into $\mathbf{Z}^l \in \mathbb{R}^{(B \times D) \times (N_{\text{obs}} + N_{\text{int}}) \times d_{\text{embed}}} = [\mathbf{Z}_{\text{obs}}^l, \mathbf{Z}_{\text{int}}^l]$. This strategy has a reduced computational cost of $\mathcal{O}(N_{\text{obs}}^2 + N_{\text{obs}} N_{\text{int}})$ and is the strategy we use for the results in the main paper.

**Attention among nodes** The output of the attention among samples at layer $l$ $\mathbf{Z}^l \in \mathbb{R}^{(B \times D) \times (N_{\text{obs}} + N_{\text{int}}) \times d_{\text{embed}}}$ is then fed into the next stage: attention among nodes. We first reshape the data $\mathbf{Z}^l \in \mathbb{R}^{(B \times D) \times (N_{\text{obs}} + N_{\text{int}}) \times d_{\text{embed}}} \rightarrow \mathbf{Z}^{l'} \in \mathbb{R}^{(B \times (N_{\text{obs}} + N_{\text{int}})) \times D \times d_{\text{embed}}}$, and then perform MHSA between the nodes:

$$\mathbf{Z}^{l+1} = \text{MHSA}(\mathbf{Z}^{l'})$$

This is then reshaped back into $\mathbf{Z}^{l+1} \in \mathbb{R}^{B \times (N_{\text{obs}} + N_{\text{int}}) \times D \times d_{\text{embed}}}$, and then split into the observational and interventional data representations that are fed into layer $l + 1$: $\mathbf{Z}_{\text{obs}}^{l+1} \in \mathbb{R}^{B \times N_{\text{obs}} \times D \times d_{\text{embed}}}$ and $\mathbf{Z}_{\text{int}}^{l+1} \in \mathbb{R}^{B \times N_{\text{int}} \times D \times d_{\text{embed}}}$.

**MACE Decoder** We parameterise the output distribution of the NP as a Mixture of Gaussians (MoG) with $N_{\text{comp}}$ components. The NP outputs the mean, standard deviation and weight corresponding to each component for each interventional query $\{x_j^n\}_{n=1}^{N_{\text{int}}}$: $\{\boldsymbol{\mu}, \boldsymbol{\sigma}, \mathbf{w}\}(x_j^n) := \{\mu_k(x_j^n), \sigma_k(x_j^n), w_k(x_j^n)\}_{k=1}^{N_{\text{comp}}}$. These are computed based on the outcome interventional representation from the final layer of the MACE Transformer Encoder. More specifically, the input to the decoder is $\mathbf{Z}_{\text{int},i}^L \in \mathbb{R}^{N_{\text{int}} \times d_{\text{embed}}}$. This is then passed through a two-layer MLP of hidden size $d_{\text{emb}}$, followed by an activation function

$$\mathbf{z}_{\text{out}} = \text{activation}(\text{MLP}(\mathbf{Z}_{\text{int},i}^L))$$

Finally, we use linear layers to project the embedding $\mathbf{z}_{\text{out}} \in \mathbb{R}^{N_{\text{int}} \times d_{\text{embed}}}$ to the parameters of a mixture of $N_{\text{comp}}$ Gaussian components:

$$\boldsymbol{\mu} = \text{Linear}_{\text{mean}}(\mathbf{z}_{\text{out}}) \in \mathbb{R}^{N_{\text{int}} \times N_{\text{comp}}}$$
$$\text{pre-}\boldsymbol{\sigma} = \text{Linear}_{\text{std}}(\mathbf{z}_{\text{out}}) \in \mathbb{R}^{N_{\text{int}} \times N_{\text{comp}}}$$
$$\text{pre-}\mathbf{w} = \text{Linear}_{\text{weight}}(\mathbf{z}_{\text{out}}) \in \mathbb{R}^{N_{\text{int}} \times N_{\text{comp}}}.$$

We then apply element-wise transforms to obtain valid parameters:

$$\boldsymbol{\sigma} = \text{softplus}(\text{pre-}\boldsymbol{\sigma}) \qquad \mathbf{w} = \text{softmax}(\text{pre-}\mathbf{w}),$$

with the softmax being applied along the component dimension.

**Loss**   The output parameters are then used to evaluate the per-dataset loss of the MACE-TNP, which, as shown in section 4 requires the evaluation of the log-posterior interventional distribution of the MoG. We restate the equation of the loss presented in section 4 for completeness:

$$\mathcal{L}_\theta(\mathbf{x}_i, \{\boldsymbol{\mu}, \boldsymbol{\sigma}, \mathbf{w}\}(\mathbf{x}_j)) = \sum_{n=1}^{N_{\text{int}}} \log p_\theta(x_i^n \,|\, \text{do}(x_j^n), \mathcal{D}) = \sum_{n=1}^{N_{\text{int}}} \log \left( \sum_{k=1}^{N_{\text{comp}}} w_k(x_j^n) \cdot \mathcal{N}(x_i^n \,|\, \mu_k(x_j^n), \sigma_k^2(x_j^n)) \right) \quad (12)$$

where $\mathcal{N}(x|\mu, \sigma)$ represents the Gaussian distribution with mean $\mu$ and standard deviation $\sigma$.

# B   Data Generation

We provide in fig. 5 a diagram showing how we sample training data from a specified Bayesian Causal Model to infer its posterior interventional distribution (see discussion in section 4).

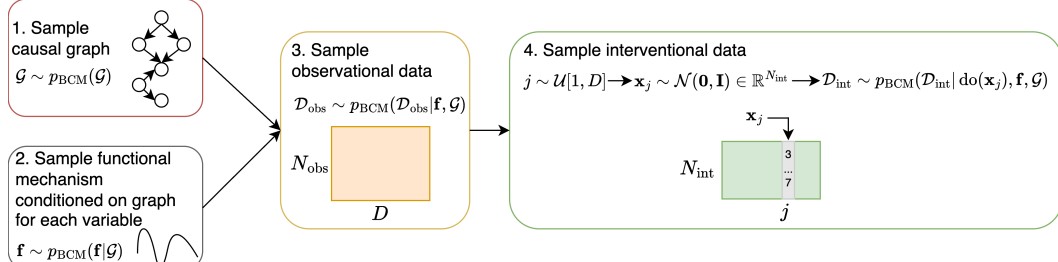

Figure 5: Overview of the data generation process. We first sample a graph $\mathcal{G}$, and a functional mechanism (conditioned on the sampled graph) for each of the $D$ nodes in the dataset. These are then used to draw $N_{\text{obs}}$ observational samples. To construct the interventional dataset, we first randomly sample a node to intervene upon $j$, draw $N_{\text{int}}$ intervention values $\mathbf{x}_j \sim \mathcal{N}(\mathbf{0}, \mathbf{I})$, and set the values of node $j$ to be $\mathbf{x}_j$. We then drawn $N_{\text{int}}$ samples of each node to form an interventional dataset $\mathcal{D}_{\text{int}}$.

## B.1   Two-node Linear Gaussian Models

The data generation details for the two-node linear Gaussian experiments from section 5.1 and the derivations of the posterior interventional distribution are explained in this section.

We examine the basic scenario involving $n$ independent and identically distributed (i.i.d.) random vectors, each consisting of two components, defined as $X^i := [X_1^i, X_2^i]^T$ for $i \in \{1, 2, \ldots, n\}$. Let the observed dataset be denoted by $\mathcal{D}_{\text{obs}} := \{X^1, X^2, \ldots, X^n\}$. For the sake of notational simplicity, we drop the subscript BCM from $p_{\text{BCM}}$ in eq. (2) throughout the subsequent proofs. In this setting, where the random vectors are composed of only two nodes $(X_1, X_2)$, there exist three distinct possible structural SCMs:

$$\mathcal{G}_1 := \begin{bmatrix} 0 & 0 \\ 1 & 0 \end{bmatrix} : X_1 = wX_2 + U_1 \text{ and } X_2 = U_2 \quad (13)$$

$$\mathcal{G}_2 := \begin{bmatrix} 0 & 1 \\ 0 & 0 \end{bmatrix} : X_1 = U_1 \text{ and } X_2 = wX_1 + U_2 \quad (14)$$

$$\mathcal{G}_3 := \begin{bmatrix} 0 & 0 \\ 0 & 0 \end{bmatrix} : X_1 = U_1 \text{ and } X_2 = U_2 \quad (15)$$

We consider two models, one where the causal graph is identifiable (appendix B.1.1) and one where it is not identifiable (appendix B.1.2).

### B.1.1  Identifiable Case

We begin with the case where the error terms $U_1$ and $U_2$ are Gaussian distributed and the noise variances of $U_1$ and $U_2$ are equal and known—a setting shown to be identifiable in Peters and Bühlmann [52]. Fixing $\sigma^2, \sigma_w^2 \in \mathbb{R}^+$, we consider the following hierarchical model:

$$\mathcal{G} \sim \mathcal{U}\{\mathcal{G}_1, \mathcal{G}_2, \mathcal{G}_3\}, \quad U_i \sim \mathcal{N}(0, \sigma^2) \quad \text{for } i = 1, 2$$
$$w \sim \mathcal{N}(0, \sigma_w^2) \quad \text{if} \quad \mathcal{G} \in \{\mathcal{G}_1, \mathcal{G}_2\}.$$

which induces the following joint distribution:

$$p(X, w, \mathcal{G}) = p(X|w, \mathcal{G})p(w|\mathcal{G})p(\mathcal{G}) \text{ if } \mathcal{G} \in \{\mathcal{G}_1, \mathcal{G}_2\} \text{ or} \tag{16}$$

$$p(X, \mathcal{G}_3) = p(X|\mathcal{G}_3)p(\mathcal{G}_3), \quad \text{otherwise.} \tag{17}$$

We show below that the above models can be identified by the posterior.

**Theorem B.1.** *Let* $\mathcal{D}_{obs} := \{X^{(1)}, X^{(2)}, \ldots, X^{(n)}\}$ *be i.i.d. observations generated by one of the simple models described in eqs. (13) to (15). The posterior over the graphs* $[p(\mathcal{G}_1|\mathcal{D}_{obs}), p(\mathcal{G}_2|\mathcal{D}_{obs}), p(\mathcal{G}_3|\mathcal{D}_{obs})]$ *is*

$$\frac{1}{c}\left[\frac{\sigma}{\sqrt{(\sigma_w^2 S_2 + \sigma^2)}} \exp\left(\frac{\sigma_w^2}{2\sigma^2}\frac{S_{12}^2}{\sigma_w^2 S_2 + \sigma^2}\right), \frac{\sigma}{\sqrt{(\sigma_w^2 S_1 + \sigma^2)}} \exp\left(\frac{\sigma_w^2}{2\sigma^2}\frac{S_{12}^2}{\sigma_w^2 S_1 + \sigma^2}\right), 1\right],$$

*where $c$ is a constant of normalisation and*

$$S_{12} := \sum_{i=1}^n X_1^i X_2^i, \quad S_1 := \sum_{i=1}^n X_1^{i2}, \quad S_2 := \sum_{i=1}^n X_2^{i2}. \tag{18}$$

*The posterior interventional distribution is a mixture of 2 Gaussian distributions*

$$p(X_1 = y|\text{do}(X_2 = x), \mathcal{D}_{obs}) = p(\mathcal{G}_1|\mathcal{D}_{obs})\mathcal{N}\left(y|\mu_{1-2}(x), \sigma_{1-2}^2(x)\right) + (1 - p(\mathcal{G}_1|\mathcal{D}_{obs}))\mathcal{N}(y|0, \sigma^2),$$

*with*

$$\mu_{1-2}(x) := \frac{\sigma_w^2 S_{12}}{\sigma_w^2 S_2 + \sigma^2} \cdot x \quad \text{and} \quad \sigma_{1-2}^2(x) := \sigma^2\left(1 + \frac{\sigma_w^2 x^2}{\sigma_w^2 S_2 + \sigma^2}\right). \tag{19}$$

**Proof:** First, we find the full conditional distribution, $p(w|\mathcal{G}, \mathcal{D}_{\text{obs}})$ and the posterior distribution over the DAG models, $p(\mathcal{G}|\mathcal{D}_{\text{obs}})$. Following Bayes' rule we have

$$p(w|\mathcal{G}_1, \mathcal{D}_{\text{obs}}) \propto p(w|\mathcal{G}_1)p(\mathcal{G}_1)\prod_{i=1}^n p(X_i|\mathcal{G}_1, w) \propto p(w|\mathcal{G}_1)\prod_{i=1}^n p(X_i|\mathcal{G}_1, w)$$

$$\propto \exp\left(-\frac{w^2}{2\sigma_w^2} - \frac{1}{2\sigma^2}\sum_{i=1}^n \left(X_1^i - wX_2^i\right)^2\right)$$

and by completing the square we obtain

$$p(w|\mathcal{G}_1, \mathcal{D}_{\text{obs}}) = \mathcal{N}\left(w\left|\frac{\sigma^2 S_{12}}{\sigma_w^2 S_2 + \sigma_1^2}, \frac{\sigma_1^2 \sigma_w^2}{\sigma_w^2 S_2 + \sigma_1^2}\right.\right), \tag{20}$$

with $S_{12}$ and $S_2$ being defined in eq. (18).

Similarly, conditioned on the $\mathcal{G}_2$ model, the full conditional distribution is again Gaussian

$$p(w|\mathcal{G}_2, \mathcal{D}_{\text{obs}}) = \mathcal{N}\left(w \left| \frac{\sigma^2 S_{12}}{\sigma_w^2 S_1 + \sigma^2}, \frac{\sigma^2 \sigma_w^2}{\sigma_w^2 S_1 + \sigma^2}\right.\right). \tag{21}$$

Using eq. (20) and eq. (21), the posterior for the $\mathcal{G}$ is

$$p(\mathcal{G}_1|\mathcal{D}_{\text{obs}}) \propto \int p(w)p(\mathcal{G}_1) \prod_{i=1}^{n} p(X^i|\mathcal{G}_1, w)dw$$

$$\propto \exp\left(-\frac{S_1 + S_2}{2\sigma^2}\right) \frac{\sigma}{\sqrt{(\sigma_w^2 S_2 + \sigma^2)}} \exp\left(\frac{\sigma_w^2}{2\sigma^2} \frac{S_{12}^2}{\sigma_w^2 S_2 + \sigma^2}\right) \tag{22}$$

$$p(\mathcal{G}_2|\mathcal{D}_{\text{obs}}) \propto \int p(w)p(\mathcal{G}_2) \prod_{i=1}^{n} p(X_i|\mathcal{G}_2, w)dw$$

$$\propto \exp\left(-\frac{S_1 + S_2}{2\sigma^2}\right) \frac{\sigma}{\sqrt{(\sigma_w^2 S_1 + \sigma^2)}} \exp\left(\frac{\sigma_w^2}{2\sigma^2} \frac{S_{12}^2}{\sigma_w^2 S_1 + \sigma^2}\right) \tag{23}$$

$$p(\mathcal{G}_3|\mathcal{D}_{\text{obs}}) \propto p(\mathcal{G}_3) \prod_{i=1}^{n} p(X_i|\mathcal{G}_3) \propto \exp\left(-\frac{S_1 + S_2}{2\sigma^2}\right). \tag{24}$$

Next, the posterior interventional distribution is

$$p(X_1|\text{do}(X_2 = x), \mathcal{D}_{\text{obs}}) = p(X_1, \mathcal{G}_3|\text{do}(x), \mathcal{D}_{\text{obs}}) + \sum_{\mathcal{G} \in \{\mathcal{G}_1, \mathcal{G}_2\}} \int p(X_1, \mathcal{G}, w|\text{do}(x), \mathcal{D}_{\text{obs}})dw$$

$$= p(X_1|\text{do}(x), \mathcal{G}_3, \mathcal{D}_{\text{obs}})p(\mathcal{G}_3|\mathcal{D}_{\text{obs}}) + \sum_{\mathcal{G} \in \{\mathcal{G}_1, \mathcal{G}_2\}} p(\mathcal{G}|\mathcal{D}_{\text{obs}}) \int p(X_1|\text{do}(x), \mathcal{G}, w)p(w|\mathcal{G}, \mathcal{D}_{\text{obs}})dw.$$

Conditioned on the model graphs, the interventional distributions are

$$p(X_1 = y|\text{do}(X_2 = x), \mathcal{G}_1, w) = \mathcal{N}(y|wx, \sigma^2), \quad p(X_1 = y|\text{do}(X_2 = x), \mathcal{G}_2, w) = \mathcal{N}(y|0, \sigma^2)$$
$$p(X_1 = y|\text{do}(X_2 = x), \mathcal{G}_3) = \mathcal{N}(y|0, \sigma^2).$$

Then, we have

$$\int p(X_1 = y|\text{do}(X_2 = x), \mathcal{G}_2, w)p(w|\mathcal{G}_2, \mathcal{D}_{\text{obs}})dw = \int \mathcal{N}(y|0, \sigma^2)p(w|\mathcal{G}_2, \mathcal{D}_{\text{obs}})dw = \mathcal{N}(y|0, \sigma^2),$$

$$\int p(X_1 = y|\text{do}(X_2 = x), \mathcal{G}_3, w)p(w|\mathcal{G}_3, \mathcal{D}_{\text{obs}})dw = \int \mathcal{N}(y|0, \sigma^2)p(w|\mathcal{G}_3, \mathcal{D}_{\text{obs}})dw = \mathcal{N}(y|0, \sigma^2),$$

and

$$\int p(X_1 = y|\text{do}(X_2 = x), \mathcal{G}_1, w)p(w|\mathcal{G}_1, \mathcal{D}_{\text{obs}})dw = \int \mathcal{N}(y|wx, \sigma^2)p(w|\mathcal{G}_1, \mathcal{D}_{\text{obs}})dw$$

$$= \mathcal{N}\left(y \left| \frac{\sigma_w^2 S_{12}}{\sigma_w^2 S_2 + \sigma^2} \cdot x, \sigma^2\left(1 + \frac{\sigma_w^2 x^2}{\sigma_w^2 S_2 + \sigma^2}\right)\right.\right).$$

Hence, the interventional distribution is simply a mixture of 2 Gaussian distributions

$$p(X_1 = y|\text{do}(X_2 = x), \mathcal{D}_{\text{obs}}) = p(\mathcal{G}_1|\mathcal{D}_{\text{obs}})\mathcal{N}\left(y|\mu_{1-2}(x), \sigma_{1-2}^2(x)\right) + (1 - p(\mathcal{G}_1|\mathcal{D}_{\text{obs}}))\mathcal{N}(y|0, \sigma^2),$$

with $p(\mathcal{G}_1|\mathcal{D}_{\text{obs}})$ calculated in eqs. (22) to (24) and $\mu_{1-2}(x), \sigma_{1-2}^2(x)$ defined in eq. (34).

Similarly, the next result easily follows

$$p(X_2 = y|\text{do}(X_1 = x), \mathcal{D}_{\text{obs}}) = p(\mathcal{G}_2|\mathcal{D}_{\text{obs}})\mathcal{N}\left(y|\mu_{2-1}(x), \sigma_{2-1}^2(x)\right) + (1 - p(\mathcal{G}_2|\mathcal{D}_{\text{obs}}))\mathcal{N}(y|0, \sigma^2),$$

with $p(\mathcal{G}_2|\mathcal{D}_{\text{obs}})$ calculated in eq. (23) and

$$\mu_{2-1}(x) := \frac{\sigma_w^2 S_{12}}{\sigma_w^2 S_1 + \sigma^2} \cdot x \quad \text{and} \quad \sigma_{2-1}^2(x) := \sigma^2\left(1 + \frac{\sigma_w^2 x^2}{\sigma_w^2 S_1 + \sigma^2}\right).$$

$\square$

**Remark:** It can be shown that if $\mathcal{D}_{\text{obs}}$ is generated by one of the models presented in eqs. (13) to (15), then the posterior distribution $p(\mathcal{G} \mid \mathcal{D}_{\text{obs}})$ asymptotically concentrates around the true data-generating structure $\mathcal{G}^*$ [12, 65, 14]. Consequently, in the infinite data limit, the posterior interventional distribution converges to a Gaussian distribution whose mean and variance depend on the intervened node and the true underlying causal mechanism $\mathcal{G}^*$.

### B.1.2 Non-identifiable Case

Second, we consider the errors' variances to be unknown while keeping the same SCMs described in eqs. (13) to (15). Therefore, we place priors on these extra parameters as well chosen such that the model is not identifiable [22]. We propose the following hierarchical model for fixed $\alpha > \frac{1}{2}$, and $\beta, \eta > 0$

$$\mathcal{G} \sim \mathcal{U}\{\mathcal{G}_1, \mathcal{G}_2, \mathcal{G}_3\}$$

If $\mathcal{G} = \mathcal{G}_1$, then $\tau_1^2 \sim InvGamma(\alpha, \beta), \tau_2^2 \sim InvGamma(\alpha - \frac{1}{2}, \beta), w \sim \mathcal{N}(0, \eta\tau_1^2)$

If $\mathcal{G} = \mathcal{G}_2$, then $\tau_1^2 \sim InvGamma(\alpha - \frac{1}{2}, \beta), \tau_2^2 \sim InvGamma(\alpha, \beta), w \sim \mathcal{N}(0, \eta\tau_2^2)$

If $\mathcal{G} = \mathcal{G}_3$, then $\tau_1^2 \sim InvGamma(\alpha - \frac{1}{2}, \beta), \tau_2^2 \sim InvGamma(\alpha, \beta)$.

Then, this hierarchical model introduces the following joint distributions

If $\mathcal{G} = \mathcal{G}_1$, then $p(X, w, \tau_1^2, \tau_2^2, \mathcal{G}_1) = p(X|w, \tau_1^2, \tau_2^2, \mathcal{G}_1)p(w|\mathcal{G}_1, \tau_1^2)p(\tau_2^2)p(\mathcal{G}_1)$

If $\mathcal{G} = \mathcal{G}_2$, then $p(X, w, \tau_1^2, \tau_2^2, \mathcal{G}_2) = p(X|w, \tau_1^2, \tau_2^2, \mathcal{G}_2)p(w|\mathcal{G}_2, \tau_2^2)p(\tau_1^2)p(\mathcal{G}_2)$

If $\mathcal{G} = \mathcal{G}_3$, then $p(X, \tau_1^2, \tau_2^2, \mathcal{G}_3) = p(X|\tau_1^2, \tau_2^2, \mathcal{G}_3)p(\tau_1^2)p(\tau_2^2)p(\mathcal{G}_3)$.

For completeness, we show below that the above priors result in the same posterior for graphs in the same Markov equivalence class — $\mathcal{G}_1$ and $\mathcal{G}_2$. We begin by recalling a simple result before stating the main theorem of this subsection.

**Lemma 1.** *For any $\nu > 0$ and $A, B, C \in \mathbb{R}$ such that $CA^2 > B$ we have*

$$\int_{-\infty}^{\infty} \frac{dx}{(A^2x^2 - 2Bx + C)^{\frac{\nu+1}{2}}} = \sqrt{\pi} \frac{\Gamma(\frac{\nu}{2})}{\Gamma(\frac{\nu+1}{2})} \times \frac{(A^2)^{\frac{\nu-1}{2}}}{(CA^2 - B^2)^{\frac{\nu}{2}}},$$

*where $\Gamma(\cdot)$ is the usual Gamma-function.*

PROOF: Completing the square in the dominator we have

$$\int_{-\infty}^{\infty} \frac{dx}{(A^2x^2 - 2Bx + C)^{\frac{\nu+1}{2}}} = \left(\frac{A^2}{CA^2 - B^2}\right)^{\frac{\nu+1}{2}} \int_{-\infty}^{\infty} \frac{dx}{\left(\frac{A^4}{CA^2 - B^2}\left(x - \frac{B}{A^2}\right)^2 + 1\right)^{\frac{\nu+1}{2}}}.$$

Next, we recall the probability density function (pdf) of a shifted and scaled version of the standard student-t distribution (i.e. $Z = \mu + \sigma T$, with $T \sim t(\nu)$)

$$f_Z(z) = \frac{\Gamma(\frac{\nu+1}{2})}{\Gamma(\frac{\nu}{2})\sqrt{\pi}}\left(1 + \frac{1}{\nu}\left(\frac{z-\mu}{\sigma^2}\right)^2\right)^{-\frac{\nu+1}{2}}.$$

Then, matching the terms gives the desired result. $\qquad\square$

**Theorem B.2.** *Let $\mathcal{D}_{obs} := \{X^{(1)}, X^{(2)}, \ldots, X^{(n)}\}$ be i.i.d. observations generated by one of the simple models described in eqs. (13) to (15). The posterior over the graphs $[p(\mathcal{G}_1|\mathcal{D}_{obs}), p(\mathcal{G}_2|\mathcal{D}_{obs}), p(\mathcal{G}_3|\mathcal{D}_{obs})]$ is*

$$\frac{1}{c}\left[\frac{(S_2^\eta)^{(\nu-1)/2}}{(S_2^\beta)^{(\nu-1)/2}[S_1^\beta S_2^\eta - S_{12}^2]^{\nu/2}}, \frac{(S_1^\eta)^{(\nu-1)/2}}{(S_1^\beta)^{(\nu-1)/2}[S_2^\beta S_1^\eta - S_{12}^2]^{\nu/2}}, \frac{1}{(S_1^\beta)^{(\nu-1)/2}(S_2^\beta)^{(\nu/2)}}\right],$$

*where $c$ is the constant of normalisation, $\nu := 2\alpha + n$ and*

$$S_i^\eta := S_i + \frac{1}{\eta}, \ S_i^\beta := S_i + 2\beta \ for \ i \in \{1, 2\}, \tag{25}$$

*with $S_1, S_2$ and $S_{12}$ defined in eq. (18). The posterior interventional distribution is a mixture of 2 shifted and scaled Student-t distributions*

$$p(X_1 = y|\mathrm{do}(X_2 = x), \mathcal{D}_{obs}) = p(\mathcal{G}_1|\mathcal{D}_{obs})t(\nu, \mu_{1-2}(x), \sigma_{1-2}(x)) + (1 - p(\mathcal{G}_1|\mathcal{D}_{obs}))t\left(\nu, 0, \sqrt{\frac{S_1^\beta}{\nu-1}}\right)$$

*with*

$$\mu_{1-2}(x) := \frac{xS_{12}}{S_2^\eta} \quad and \quad \sigma_{1-2}^2(x) := \frac{S_2^\eta(S_1^\beta S_2^\eta - S_{12}^2) + x^2(S_1^\beta S_2^\eta - S_{12}^2)}{(S_2^\eta)^2\nu}. \tag{26}$$

**Proof:** Similar to the case presented in Appendix B.1.1, we start by deriving the posterior over the three models described above and use the same definitions from eq. (18).

$$p(\mathcal{G}_1|\mathcal{D}_{\mathrm{obs}}) \propto \int p(\mathcal{D}_{\mathrm{obs}}|\mathcal{G}_1, w, \tau_1^2, \tau_2^2)p(w|\tau_1^2)p(\tau_1^2)p(\tau_2^2)d(\tau_1^2) \ d(\tau_2^2) \ dw$$

$$= \int \frac{1}{\left(\sqrt{2\pi\tau_1^2}\right)^n}\exp\left(-\frac{1}{2\tau_1^2}\sum_{i=1}^n(X_1^i - wX_2^i)^2\right)\frac{1}{\left(\sqrt{2\pi\tau_2^2}\right)^n}\exp\left(-\frac{1}{2\tau_2^2}S_2\right)\frac{1}{\sqrt{2\pi\eta\tau_1^2}}$$

$$\times \exp\left(-\frac{w^2}{2\eta\tau_1^2}\right)\frac{\beta^{2\alpha-\frac{1}{2}}}{\Gamma(\alpha)\Gamma(\alpha-\frac{1}{2})}(\tau_1^2)^{-\alpha-1}\exp\left(-\frac{\beta}{\tau_1^2}\right)(\tau_2^2)^{-\alpha-1/2}\exp\left(-\frac{\beta}{\tau_2^2}\right)d(\tau_1^2) \ d(\tau_2^2) \ dw$$

$$= \frac{1}{\sqrt{\eta}}\frac{1}{(2\pi)^{n+1/2}}\frac{\beta^{2\alpha-1/2}}{\Gamma(\alpha)\Gamma(\alpha-\frac{1}{2})}\frac{\Gamma(\alpha+\frac{n-1}{2})}{\left(\beta+\frac{S_2}{2}\right)^{(\nu-1)/2}}\int_{-\infty}^\infty \frac{\Gamma(\alpha+\frac{n+1}{2})}{\left(\beta+\frac{\sum_{i=1}^n(X_1^i-wX_2^i)^2}{2}+\frac{w^2}{2\eta}\right)^{\alpha+\frac{n+1}{2}}}dw$$

$$= \frac{(2\beta)^{2\alpha-\frac{1}{2}}\Gamma(\alpha+\frac{n-1}{2})\Gamma(\alpha+\frac{n}{2})}{\sqrt{\eta}\pi^n\Gamma(\alpha)\Gamma(\alpha-\frac{1}{2})}\cdot\frac{(1/\eta+S_2)^{(\nu-1)/2}}{(2\beta+S_2)^{(\nu-1)/2}}\cdot\frac{1}{[(2\beta+S_1)(1/\eta+S_2)-S_{12}^2]^{\alpha+n/2}}$$

$$= \frac{(2\beta)^{2\alpha-\frac{1}{2}}\Gamma(\alpha+\frac{n-1}{2})\Gamma(\alpha+\frac{n}{2})}{\sqrt{\eta}\pi^n\Gamma(\alpha)\Gamma(\alpha-\frac{1}{2})}\cdot\left(\frac{S_2^\eta}{S_2^\beta}\right)^{(\nu-1)/2}\cdot\frac{1}{(S_1^\beta S_2^\eta - S_{12}^2)^{\nu/2}}, \tag{27}$$

where in the last step we used the result of Lemma 1. We note that the conditions in the lemma are fulfilled as

$$(2\beta+S_1)(1/\eta+S_2) > S_1S_2 \geq \left(S_{12}\right)^2, \tag{28}$$

where we used the fact that $\beta, \eta > 0$ and the Cauchy-Schwartz inequality in the second step. Similarly, we find the posterior for the second model, $\mathcal{G}_2$

$$p(\mathcal{G}_2|\mathcal{D}_{\text{obs}}) \propto \frac{(2\beta)^{2\alpha-\frac{1}{2}}\Gamma(\alpha+\frac{n-1}{2})\Gamma(\alpha+\frac{n}{2})}{\sqrt{\eta}\pi^n\Gamma(\alpha)\Gamma(\alpha-\frac{1}{2})} \cdot \left(\frac{S_1^\eta}{S_1^\beta}\right)^{(\nu-1)/2} \cdot \frac{1}{(S_2^\beta S_1^\eta - S_{12}^2)^{\nu/2}}$$

and for the third model, $\mathcal{G}_3$

$$
\begin{aligned}
p(\mathcal{G}_3|\mathcal{D}_{\text{obs}}) &\propto \int p(\mathcal{D}_{\text{obs}}|\mathcal{G}_3, \tau_1^2, \tau_2^2)p(\tau_1^2)p(\tau_2^2)d(\tau_1^2)\ d(\tau_2^2) \\
&= \int \frac{1}{\left(\sqrt{2\pi\tau_1^2}\right)^n}\exp\left(-\frac{1}{2\tau_1^2}S_1\right)\frac{1}{\left(\sqrt{2\pi\tau_2^2}\right)^n}\exp\left(-\frac{1}{2\tau_2^2}S_2\right) \\
&\quad \times \frac{\beta^{2\alpha-1/2}}{\Gamma(\alpha)\Gamma(\alpha-\frac{1}{2})}(\tau_1^2)^{-\alpha-1/2}\exp\left(-\frac{\beta}{\tau_1^2}\right)(\tau_2^2)^{-\alpha-1}\exp\left(-\frac{\beta}{\tau_2^2}\right)d(\tau_1^2)\ d(\tau_2^2) \quad (29) \\
&= \frac{(2\beta)^{2\alpha-1/2}\Gamma(\alpha+\frac{n-1}{2})\Gamma(\alpha+\frac{n}{2})}{\pi^n\Gamma(\alpha)\Gamma(\alpha-\frac{1}{2})} \cdot \frac{1}{(S_1^\beta)^{(\nu-1)/2}(S_2^\beta)^{\nu/2}}. \quad (30)
\end{aligned}
$$

Conditioned on the models, the interventional distributions are

$$p(X_1 = y|\text{do}(x), \mathcal{G}_1, w, \tau_1^2) := p(X_1 = y|\text{do}(X_2 = x), \mathcal{G}_1, w, \tau_1^2) = \mathcal{N}(y|wx, \tau_1^2),$$
$$p(X_1 = y|\text{do}(x), \mathcal{G}_2, w, \tau_1^2) = p(X_1 = y|\text{do}(X_2 = x), \mathcal{G}_2, w, \tau_1^2) = \mathcal{N}(y|0, \tau_1^2),$$
$$p(X_1 = y|\text{do}(x), \mathcal{G}_3, \tau_1^2) := p(X_1 = y|\text{do}(X_2 = x), \mathcal{G}_3, \tau_1^2) = \mathcal{N}(y|0, \tau_1^2).$$

Then, the posterior interventional distribution, $p(X_1|\text{do}(x), \mathcal{D}_{\text{obs}})$, is

$$\sum_{\mathcal{G}\in\{\mathcal{G}_1,\mathcal{G}_2\}} p(\mathcal{G}|\mathcal{D}_{\text{obs}})\int p(X_1 = y|\text{do}(x), \mathcal{G}, w, \tau_1^2)p(w, \tau_1^2, \tau_2^2|, \mathcal{G}, \mathcal{D}_{\text{obs}})\ d(\tau_1^2)\ d(\tau_2^2)\ dw$$

$$+ p(\mathcal{G}_3|\mathcal{D}_{\text{obs}})\int p(X_1 = y|\text{do}(x), \mathcal{G}_3, \tau_1^2)p(\tau_1^2, \tau_2^2|, \mathcal{G}_3, \mathcal{D}_{\text{obs}})\ d(\tau_1^2)\ d(\tau_2^2).$$

First, we find the last term

$$\int p(X_1 = y|\text{do}(X_2 = x), \mathcal{G}_3, \tau_1^2)p(\tau_1^2, \tau_2^2|, \mathcal{G}_3, \mathcal{D}_{\text{obs}})\ d(\tau_1^2)\ d(\tau_2^2) \propto$$

$$\int \frac{1}{\sqrt{2\pi\tau_1^2}}\exp\left(-\frac{y^2}{2\tau_1^2}\right)p(\mathcal{D}_{\text{obs}}|\mathcal{G}_3, \tau_1^2, \tau_2^2)p(\tau_1^2)p(\tau_2^2)d(\tau_1^2)\ d(\tau_2^2) \propto$$

$$\propto \frac{1}{(2\beta + S_1 + y^2)^{\alpha+\frac{n-1}{2}}} \propto \frac{1}{(1 + \frac{y^2}{S_1^\beta})^{\alpha+\frac{n-1}{2}}}, \quad (31)$$

which is a scaled Student-t distribution with $\nu = 2\alpha + n$ degrees of freedom and $\sigma^2 = \frac{S_1^\beta}{\nu-1}$. Computing the next integral follows the same pattern presented in eq. (30)

$$\int p(X_1 = y|\text{do}(X_2 = x), \mathcal{G}_2, w, \tau_1^2)p(w, \tau_1^2, \tau_2^2|, \mathcal{G}_2, \mathcal{D}_{\text{obs}})\ d(\tau_1^2)\ d(\tau_2^2)\ dw \propto$$

$$\int \frac{1}{\sqrt{2\pi\tau_1^2}}\exp\left(-\frac{y^2}{2\tau_1^2}\right)p(\mathcal{D}_{\text{obs}}|w, \tau_1^2, \tau_2^2)p(w|\tau_2^2)p(\tau_1^2)p(\tau_2^2)\ d(\tau_1^2)\ d(\tau_2^2)\ dw \propto$$

$$\frac{(S_1 + \frac{1}{\eta})^{\alpha+\frac{n-1}{2}}}{(2\beta + S_1 + y^2)^{\alpha+\frac{n-1}{2}}} \cdot \frac{1}{\left[(2\beta + S_2)(S_1 + \frac{1}{\eta}) - (S_{12})^2\right]} \propto \frac{1}{(1 + \frac{y^2}{S_1^\beta})^{\alpha+\frac{n-1}{2}}}, \quad (32)$$

which is again a scaled Student-t distribution with $\sigma^2 = \frac{S_1^\beta}{\nu-1}$ and $\nu$ degrees of freedom. Finally, using similar steps as in eq. (27), the last term is

$$\int p(X_1 = y|\mathrm{do}(X_2 = x), \mathcal{G}_1, w, \tau_1^2)p(w, \tau_1^2, \tau_2^2|, \mathcal{G}_1, \mathcal{D}_{\mathrm{obs}}) \, d(\tau_1^2) \, d(\tau_2^2) \, dw \propto$$

$$\int \frac{1}{\sqrt{2\pi\tau_1^2}} \exp\left(-\frac{(y-wx)^2}{2\tau_1^2}\right) p(\mathcal{D}_{\mathrm{obs}}|w, \tau_1^2, \tau_2^2)p(w|\tau_1^2)p(\tau_1^2)fp\tau_2^2) \, d(\tau_1^2) \, d(\tau_2^2) \, dw \propto$$

$$\frac{1}{\left[(S_2 + x^2 + \frac{1}{\eta})(2\beta + S_1 + y^2) - (S_{12} + xy)^2\right]^{\alpha + \frac{n}{2}}}, \tag{33}$$

where the denominator is always bounded away from $0$ as in eq. (28). By completing the square, we obtain another scaled and shifted student-t distribution with $\nu = 2\alpha + n$, $\mu_{1-2}(x)$ and $\sigma_{1-2}^2(x)$ defined in eq. (34). Then, combining eqs. (27) to (33) we obtain the result.

Similarly, we show

$$p(X_2 = y|\mathrm{do}(x), \mathcal{D}_{\mathrm{obs}}) = p(\mathcal{G}_2|\mathcal{D}_{\mathrm{obs}})t(\nu, \mu_{2-1}(x), \sigma_{2-1}(x)) + (1 - p(\mathcal{G}_2|\mathcal{D}_{\mathrm{obs}}))t\left(\nu, 0, \sqrt{\frac{S_2^\beta}{\nu-1}}\right)$$

with

$$\mu_{2-1}(x) := \frac{xS_{12}}{S_1^\eta} \quad \text{and} \quad \sigma_{2-1}^2(x) := \frac{S_1^\eta(S_2^\beta S_1^\eta - S_{12}^2) + x^2(S_2^\beta S_1^\eta - S_{12}^2)}{(S_1^\eta)^2\nu}. \tag{34}$$

$\square$

**Remark:** We employ asymmetric priors for $\tau_1^2$ and $\tau_2^2$ to ensure that the posterior assigns equal probability to the models $\mathcal{G}_1$ and $\mathcal{G}_2$, which belong to the same Markov equivalence class. In particular, setting $2\beta = \eta$ results in $S_1^\eta = S_1^\beta$ and $S_2^\eta = S_2^\beta$, which implies that the posterior distribution over graphs,

$$\left[p(\mathcal{G}_1 \mid \mathcal{D}_{\mathrm{obs}}), \, p(\mathcal{G}_2 \mid \mathcal{D}_{\mathrm{obs}}), \, p(\mathcal{G}_3 \mid \mathcal{D}_{\mathrm{obs}})\right],$$

takes the form:

$$\frac{1}{c}\left[\frac{1}{(S_1^\beta S_2^\beta - S_{12}^2)^{\nu/2}}, \, \frac{1}{(S_2^\beta S_1^\beta - S_{12}^2)^{\nu/2}}, \, \frac{1}{(S_1^\beta)^{(\nu-1)/2}(S_2^\beta)^{\nu/2}}\right], \tag{35}$$

where $c$ is a normalising constant.

This setup corresponds to the prior structure proposed by Geiger and Heckerman [22, Equation 12], obtained by setting the precision matrix $T$ in the Wishart distribution to the identity. As noted in their Geiger and Heckerman [22, Section 4], a change of variables transforms the Wishart prior on the covariance of $X$ into the prior used here for the weights $w$ and error variances $\tau_1^2$ and $\tau_2^2$.

Assuming a true data-generating mechanism, the posterior concentrates on its Markov equivalence class. If the true graph is $\mathcal{G}_1$ or $\mathcal{G}_2$, then $p(\mathcal{G}_1|\mathcal{D}_{\mathrm{obs}}) = p(\mathcal{G}_2|\mathcal{D}_{\mathrm{obs}}) \to 1/2$. Conversely, if $\mathcal{G}_3$ is the true graph, then $p(\mathcal{G}_3|\mathcal{D}_{\mathrm{obs}}) \to 1$[12, 65, 14].

The degrees of freedom, $\nu = \alpha + n$, grow with the sample size $n$, so the corresponding Student-$t$ distributions converge to Gaussians in the large-sample regime.

## B.2 Three-node Experiments

In the three-node experiments (section 5.3) we use two datasets with two different functional mechanisms $f_i(\cdot)$ as defined in eq. (1): one sampled from a GP prior, and one based on neural networks. In both cases, we sample Erdős–Rényi graphs with graph degree chosen uniformly from $\{1, 2, 3\}$. Following Ormaniec et al. [49], we standardise all variables upon generation.

**GP functional mechanism**   To model $f_i(\cdot)$ we use a GP with a squared exponential kernel, with a randomly sampled lengthscale for each parent set $PA_i$ of size $|PA_i|$. More specifically, we sample the lengthscale from a log- distribution $\{\lambda_p\}_{p=1}^{|PA_i|} \sim \text{Log}(-1, 1)$, followed by clipping between $\lambda_p = \text{clip}(\lambda_p, 0.1, 5)$ to ensure that a too long lengthscale does not result in independence of the variable from a parent. This defines the kernel matrix between the $n$-th and $m$-th samples as:

$$\mathbf{K}_{nm} = \exp(-(PA_i^n - PA_i^m)^T \mathbf{\Lambda}^{-1}(PA_i^n - PA_i^m)),$$

with $\mathbf{\Lambda} := \text{Diag}(\lambda_1, \ldots, \lambda_{|PA_i|})$. We then add noise with variance $\sigma^2 \sim \text{Gamma}(1, 5)$ and sample the variables as follows

$$X_i \sim \mathcal{N}(\mathbf{0}, \mathbf{K} + \sigma^2 \mathbf{I})$$

**Neural network-based functional mechanism**   We sample each variable as follows

$$\eta_i \sim \mathcal{N}(0, 1)$$
$$X_i^n \sim \text{ResNet}_\theta([PA_i^n, \eta_i]) + \sigma\epsilon,$$

where $\sigma^2 \sim \text{Gamma}(1, 10)$, $\epsilon \sim \mathcal{N}(0, 1)$. $\text{ResNet}_\theta$ is a residual neural network with a randomly sampled number of blocks $N_{\text{blocks}} \sim \mathcal{U}\{1, \ldots, 8\}$ and randomly sampled hidden dimension $d_{\text{hidden}} \sim \mathcal{U}\{2^5, 2^6, 2^7, 2^8\}$. We use the GELU [27] activation function.

### B.3   Higher-dimensional experiments

For the higher dimension experiments in section 5.4, we generate the training data for MACE-TNP as follows:

- We sample number of variables $D \sim \mathcal{U}[5, 40]$.
- We sample a type of graph, either an Erdős–Rényi graph or a scale-free graph [3].
- The density of the graph (number of edges) is sampled from $\mathcal{U}\left[\frac{D}{2}, 6D\right]$.
- For each node, we sample a functional mechanism randomly from either a GP with an additional latent variable input, or a Neural network with an additional latent variable input:
  - GP with latent: We sample a latent $\eta_i \sim \mathcal{N}(0, 1)$, and lengthscales $\{\lambda_p\}_{p=1}^{|PA_i|+1} \sim \text{Log}(-0.5, 1)$, where $PA_i$ denotes the set of parents of node index $i$. Functions are sampled from a squared expoenential kernel with $\eta_i$ included as an input and Gaussian noise added with variance $\sigma^2 \sim \text{Gamma}(1, 5)$.
  - NN with latent:

    $$\eta_i \sim \mathcal{N}(0, 1)$$
    $$X_i^n \sim \text{NN}_\theta([PA_i^n, \eta_i]) + \sigma\epsilon,$$

    where $\sigma^2 \sim \text{Gamma}(1, 10)$, $\epsilon \sim \mathcal{N}(0, 1)$. $\text{NN}_\theta$ denotes a randomly initialised neural network with 128 hidden dimensions and one hidden layer.

Using a latent as an input ensures that the final distribution is not Gaussian. Following Ormaniec et al. [49], we standardise all variables during the data generation process.

For testing for each variable size in table 2, we only generate Erdős–Rényi graphs with density $4D$. This is to test the performance of the baselines and our method in the difficult dense graph case. The rest of the data generation process is the same as the training data.

# C   Experimental Details

This section provides additional details and results for the experiments presented in Section 5.

## C.1   Architecture, training details and hardware

Throughout our experiments we use $H = 8$ attention heads, each of dimension $D_Q = D_{KV} = d_{\mathrm{model}}/8$. The MLPs used in the encoding use two layers and a hidden dimension of $d_{\mathrm{embed}} = d_{\mathrm{model}}$. Unless otherwise specified, we use a learning rate of $5 \times 10^{-4}$ with a linear warmup of $2\%$ of the total iterations, and a batch size of 32.

To train MACE-TNP, we randomise the number of observational samples $N_{\mathrm{obs}} \sim \mathcal{U}\{50, 750\}$, and set $N_{\mathrm{int}} = 1000 - N_{\mathrm{obs}}$. The training loss is evaluated on these $N_{\mathrm{int}}$ samples. For testing, we sample 500 observation points and compute the loss against 500 intervention points.

**Two-node linear Gaussian model**   We use $L = 2$ transformer encoder layers, where each transformer encoder layer involves the attention over samples, followed by attention over nodes. The model dimension is $d_{\mathrm{model}} = 128$, and feedforward width $d_{\mathrm{ff}} = 128$. We train the model for 1 epoch on 50.000 datasets and test on 100 datasets. Training takes roughly 60 minutes on a single NVIDIA GeForce RTX 2080 Ti GPU 11GB, and testing is performed in less than 5 seconds.

**Three-node experiments**   For the experiment in the main paper, we use $L = 2$ transformer encoder layers, a model dimension $d_{\mathrm{model}} = 128$, and feedforward width $d_{\mathrm{ff}} = 128$. We train the model for 2 epochs on 50.000 datasets for the GP experiment and 100.000 datasets for the NN one, and test on 100 datasets in both cases. When testing the OOD performance, we train on the union of the two datasets for 2 epochs. Training the models described in the main text required roughly $4 - 6$ hours of GPU time; however, because we ran them on a shared cluster, actual runtimes may vary with cluster utilization.

**Higher dimensional and Sachs experiments**   For the higher dimensional experiments we use $L = 4$ encoder layers. The model dimension is $d_{\mathrm{model}} = 256$ with feedforward dimension $d_{\mathrm{ff}} = 1024$. We train the model on data generated as listed in appendix B.3, with $2, 500, 000$ datasets in total. The model was trained on an NVIDAI A100 80GB GPU for 2 epochs which took roughly 20 hours. We use the model trained for the higher dimensional experiment for the Sachs experiment.

**Hardware**   For the two- and three-node experiments, we ran both training and inference on a single NVIDIA GeForce RTX 2080 Ti (11 GB) with 20 CPU cores on a shared cluster. The only exception was for our largest three-node GP and NN models (with $d_{\mathrm{model}} = 1024$), where we used a single NVIDIA RTX 6000 Ada Generation (50 GB) paired with 56 CPU cores; those models required roughly 25 GB of GPU memory. For the higher-node experiments, we used a single NVIDIA A100 80GB GPU, as well as an RTX 4090 24GB GPU.

## C.2   Additional Results

### C.2.1   Two-node Linear Gaussian Model

We study the performance of MACE-TNP in both identifiable and non-identifiable causal settings by generating data according to the models described in appendix B.1.1 and appendix B.1.2, respectively. For all experiments, we set $\sigma = \sigma_w = 1$ in the identifiable case and $\alpha = 3$, $\eta = 2\beta = 1$ in the non-identifiable case. We investigate at different interventional queries, $x$, how the NP predicted distributions compare with the analytical ones as a function of the observational sample size. For simplicity, for the first model B.1.1 we consider an NP which outputs a mixture of 2 components, while for the second model B.1.2, the NP approximates the true analytical distribution using 3 components.

The flexibility of our architecture also allows for conditional queries, multiple interventions, as well as easily incorporating interventional data to help identify causal relations. Hence, we investigate here whether providing a small number $M_{\mathrm{int}} = 5$ of true interventional samples, alongside the observational data, resolves identifiability challenges in the non-identifiable case. As already shown in fig. 3 (right) with the green line and discussed in section 5.1, we find that adding extra interventional data

does indeed lower the $\mathrm{KL}(p_{BCM}(x_i|\mathrm{do}(x_j),\mathbf{f}^*,\mathcal{G}^*)\|p_\theta(x_i|\mathrm{do}(x_j),\mathcal{D}_{\mathrm{obs}},\{x_i^n\}_{n=1}^{M_{\mathrm{int}}}))$, suggesting that even limited interventions can enhance identifiability. We also test this with an increasing number of interventional samples in fig. 6. As soon as the interventional information is rich enough ($M_{\mathrm{int}} \in \{50, 300\}$), the NP recovers the interventional disribution of the true data-generating mechanism even with little to no observational data, as indicated by the near-flat KL curves. We note that the KL divergence between two Gaussian mixtures (or between a Student-t mixture and a Gaussian mixture) lacks a closed-form expression. Therefore, we approximate it in our experiments by averaging 1000 Monte Carlo estimates of the log-density ratio.

Then, we show in fig. 7 two examples where the intervention is made at $x = 1$ for the identifiable model and at $x = 2$ for the non-identifiable model. A clear distinction is observed between the two settings: for the identifiable case, the analytical posterior interventional distribution is a mixture of two Gaussian distributions, which, at high observational sample sizes, converges to a single Gaussian (i.e. because the observational data gives information regarding the causal structure, the weight corresponding to one mode collapses to 0). In contrast, for the non-identifiable case, the posterior places equal mass on both $\mathcal{G}_1$ and $\mathcal{G}_2$, and therefore, the mixture structure persists across both regimes. In both settings, the NP-predicted distributions closely match the correct interventional distributions, with accuracy improving as the number of observational samples increases. This improvement is due to two factors. First, larger sample sizes provide the NP with more information about the underlying causal model, allowing for enhanced inference. Second, in the non-identifiable case, the posterior interventional distribution is a mixture of two Student-t distributions with a number of degrees of freedom proportional to the number of observational samples. Thus, in the high sample regime, the mixture distribution converges to a mixture of Gaussians, which is the class that parameterises the output of the NP model.

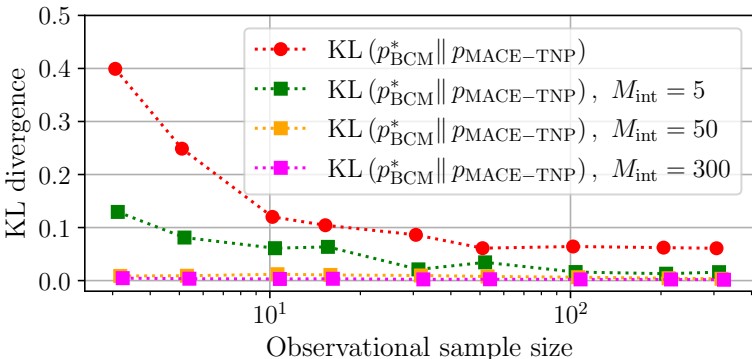

Figure 6: Average KL divergence between the interventional distribution of the true generating mechanism, $\{\mathcal{G}^*, \mathbf{f}^*\}$, and the NP-predicted distribution shown as a function of the observational sample size for the non-identifiable setting. Results are shown for various interventional sample sizes. For simplicity, we only report the medians.

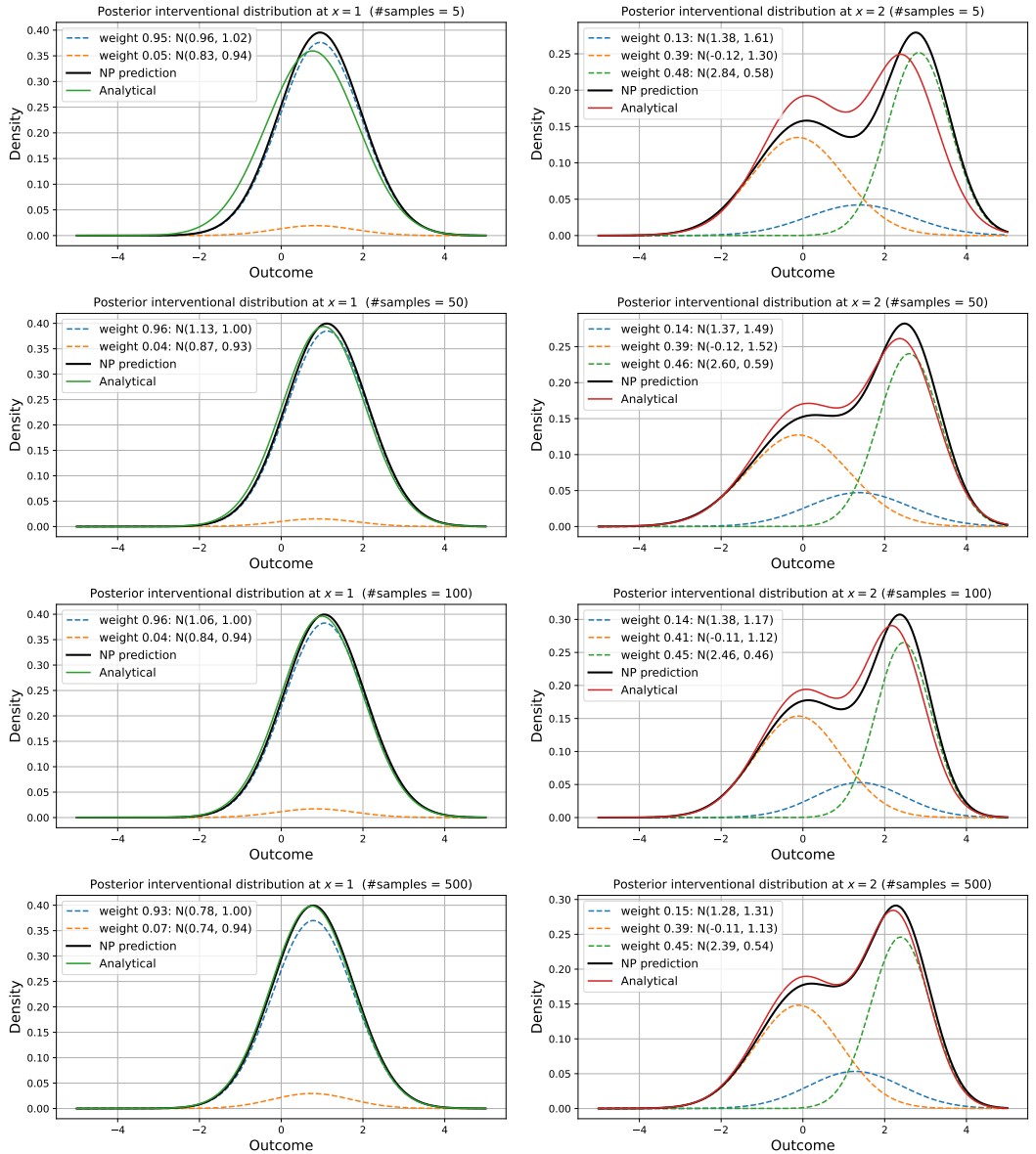

Figure 7: Fitted NP posterior interventional distributions vs. true posterior interventional distributions for identifiable (left) and non-identifiable models (right) at increasing observational sample sizes (5, 50, 100, 500).

### C.2.2   Three-node Experiments

In this section we provide additional results on the three-node experiments where we aim to address three questions: 1) between the MHSA and MHCA schemes for sample attention introduced in appendix A.2, which one performs better? 2) Does increasing the number of MoG components improve performance, and 3) How does the model performance vary with the size of the architecture?

Table 3 shows the results for the two functional mechanisms used in the three-node experiments: GP and NN-based. For each model configuration, we present four sets of results: for a model trained on GP and tested on GP (GP / GP), a model trained on NN and tested on NN (NN / NN), and for a model trained on the combination between the two datasets and tested on each of them (GP+NN / GP and GP+NN / NN). These results allow us to assess whether the influence of model architecture is consistent across the functional mechanisms. Notably, models trained on the combined GP+NN

dataset are able to match—within error—the performance of models trained specifically on either GP or NN data. This highlights the strength of the meta-learning approach: even when trained on data generated from diverse functional mechanisms, a single model can generalise effectively across both, achieving performance comparable to specialised models while also benefiting from broader prior coverage. We summarise the findings from table 3:

1. MHSA + MHCA outperforms the masked-MHSA strategy for attention over samples.

2. Increasing the number of MoG components increases the performance of MACE-TNP. There is a larger gap in performance when going from 1 to 3 mixture components, indicating the importance of allowing the model to output non-Gaussian marginal predictions. Increasing the number of components to 10 further improves performance, but the gains are not as significant.

3. Scaling up the model architecture generally leads to decreased NLPID.

4. Training a model on the combination of the two datasets (GP+NN) is able to recover—within error—the performance on both datasets.

Table 3: Results of MACE-TNP under different architectural configurations. M-SA stands for Masked-MHSA, while SA+CA indicates the MHSA+MHCA attention mechanism. For each model, the column name under NLPID indicates the training set / test set (i.e. GP+NN / GP indicates we trained the model on the GP+NN dataset and tested it on the GP one). We report the mean $\pm$ the error of the mean of the NLPID over 100 datasets.

| | | | | | NLPID ($\downarrow$) | | | |
|---|---|---|---|---|---|---|---|---|
| MoG | Attention | $d_{\text{model}}$ | $d_{\text{ff}}$ | $L$ | GP / GP | NN / NN | GP+NN / GP | GP+NN / NN |
| 1 | M-SA | 128 | 128 | 4 | $629.0 \pm 20.0$ | $664.1 \pm 16.0$ | $640.1 \pm 17.3$ | $668.3 \pm 17.2$ |
| 1 | SA+CA | 128 | 128 | 4 | $617.4 \pm 20.1$ | $664.9 \pm 16.4$ | $629.8 \pm 17.5$ | $688.0 \pm 31.5$ |
| 3 | M-SA | 128 | 128 | 4 | $581.8 \pm 21.8$ | $538.9 \pm 19.1$ | $597.6 \pm 19.9$ | $547.1 \pm 17.8$ |
| 3 | SA+CA | 128 | 128 | 4 | $569.3 \pm 23.1$ | $540.5 \pm 17.1$ | $582.1 \pm 21.6$ | $540.6 \pm 18.8$ |
| 10 | M-SA | 128 | 128 | 4 | $572.1 \pm 21.9$ | $533.2 \pm 18.3$ | $599.4 \pm 20.3$ | $531.5 \pm 19.6$ |
| 10 | SA+CA | 128 | 128 | 4 | $563.9 \pm 23.4$ | $527.9 \pm 19.8$ | $583.9 \pm 21.5$ | $531.0 \pm 19.4$ |
| 10 | SA+CA | 512 | 256 | 8 | $555.7 \pm 24.6$ | $527.0 \pm 19.1$ | $564.6 \pm 23.6$ | $532.1 \pm 18.4$ |
| 10 | SA+CA | 1024 | 256 | 8 | $558.0 \pm 23.9$ | $518.2 \pm 19.7$ | $565.6 \pm 22.3$ | $521.1 \pm 20.8$ |

Finally, table 4 summarises the results for the out-of-distribution (OOD) evaluation for the configuration presented in the main text.

Table 4: Results for the OOD two-node experiment. We show the NLPID ($\downarrow$) and report the mean $\pm$ the error of the mean over 100 datasets. Each row corresponds to a different functional mechanism used in the test set (GP / NN).

| | Training $\rightarrow$ | | |
|---|---|---|---|
| Test $\downarrow$ | GP | NN | GP+NN |
| GP | $\mathbf{563.9 \pm 23.4}$ | $678.0 \pm 10.0$ | $\mathbf{583.9 \pm 21.5}$ |
| NN | $608.3 \pm 17.3$ | $\mathbf{527.9 \pm 19.8}$ | $\mathbf{531.0 \pm 19.4}$ |

### C.2.3 Sachs Full Results

The full set of results for the Sachs dataset are shown in table 5. The MACE-TNP performs competitively with DECI, and both outperform other methods that use GPs as functional models.

Table 5: Results for the Sachs dataset [55]. We show the NLPID ($\downarrow$) and report the mean $\pm$ the error of the mean across 5 interventions and across all 10 nodes used as the outcome for each intervention. Each row corresponds to a different baseline.

|  | Sachs |
| --- | --- |
| **MACE-TNP** | $\mathbf{998.9 \pm 104.9}$ |
| **DiBS-GP** | $1417.5 \pm 186.7$ |
| **ARCO-GP** | $1400.7 \pm 208.7$ |
| **DECI** | $\mathbf{1000.9 \pm 133.5}$ |
| **NOGAM+GP** | $1763.7 \pm 297.4$ |

