# OpenReview forum: "Estimating Interventional Distributions with Uncertain Causal Graphs through Meta-Learning"
_NeurIPS.cc/2025/Conference — NeurIPS 2025 poster_

### Official Review · Reviewer_2FPW · 2025-06-29

**Clarity:** 3
**Significance:** 3
**Originality:** 3
**Rating:** 5
**Confidence:** 4

**Summary:**

This paper tackles the challenge of estimating interventional distributions when the true causal graph is unknown and must be inferred from observational data. Instead of relying on a single estimated causal structure—which can lead to overconfident or incorrect conclusions—the authors propose using meta-learning to approximate the full Bayesian model-averaged interventional distribution.

They introduce MACE-TNP (Model-Averaged Causal Estimation Transformer Neural Process), an end-to-end framework that bypasses the intractable steps of explicitly modeling posteriors over causal structures and functions. By directly learning to predict the interventional posterior, their approach avoids compounding errors and scales to more complex scenarios.

Empirical results show that MACE-TNP recovers known analytical posteriors when available, outperforms strong Bayesian and non-Bayesian baselines across diverse experiments, and offers a promising path toward scalable, meta-learning-based causal inference.

**Questions:**

Please respond to the weakness section.

**Ethical Concerns:**

["NO or VERY MINOR ethics concerns only"]

**Final Justification:**

I am reasonably satisfied with the authors' response. Some of the issues raised are common in the field and largely beyond the authors' control; therefore, I will not penalize them for these.

**Limitations:**

Yes

**Quality:**

3

**Strengths And Weaknesses:**

Strengths:
1) Addresses an important problem: Tackles the realistic and challenging scenario where causal graphs are uncertain, avoiding overconfidence from single-structure methods.
2) Principled approach: Builds on Bayesian model averaging to handle structural uncertainty in a theoretically sound way.
3) Innovative use of meta-learning: Leverages neural processes to amortize the entire causal inference pipeline, bypassing otherwise intractable computations.
4) Strong empirical validation: Demonstrates that MACE-TNP recovers known analytical posteriors and outperforms existing Bayesian and non-Bayesian baselines across increasingly complex settings.
5) Potential scalability: Shows promise for scaling to high-dimensional problems where traditional Bayesian methods become impractical.

Weaknesses:
1) Reliance on synthetic data: Most results are on synthetic datasets; it remains unclear how well the method performs on noisy, imperfect real-world data.
2) Interpretability limitations: While the method predicts interventional distributions, it offers limited insight into the underlying causal structures or mechanisms it learns, which may be important in scientific applications.
3) Assumes training distributions generalize: The success of the meta-learning approach depends on training data sufficiently representing the range of possible causal scenarios; this may limit applicability in domains with very different or unseen causal patterns.
4) Potential sensitivity: As a meta-learned model, it may be sensitive to hyperparameters or require large, well-designed training datasets to achieve robust performance.

---

> ### Author Rebuttal · Authors · 2025-07-31
>
> We thank the reviewer for their thoughtful feedback. We're glad they found our approach **principled**, our use of **meta-learning innovative**, and appreciated the **strong empirical** results showing MACE-TNP's ability to recover analytical posteriors and outperform strong baselines. We're also encouraged by their recognition of the method’s **scalability** to complex, high-dimensional settings.
>
> > Reliance on synthetic data
>
> We agree that real-world evaluation is crucial, **which is why we included experiments on the real-world Sachs dataset in Section 5.4**. However, the lack of established benchmarks with reliable interventional data remains a challenge in the community. Despite this, we are actively exploring additional real-world settings to further assess the strengths and limitations of our approach beyond synthetic benchmarks.
>
>
> > Interpretability limitations ... limited insight into the underlying causal structures or mechanisms
>
> As mentioned in our reply to reviewer RHFT, we agree this is a valid concern. We focus on accurately estimating interventional distributions, rather than recovering interpretable causal graphs or functions. However, since the model can output interventional effects for all node pairs, it is in principle possible to reconstruct the underlying causal structure from these outputs.
>
> Moreover, our framework is flexible and could be extended to **jointly estimate a posterior over causal graphs**—for example, by adding an auxiliary decoder head, as done in [1]. We see this as a promising direction for future work where interpretability is essential, and will mention it in the Conclusions section of the updated manuscript. We thank the reviewer for highlighting this point.
>
> [1] Dhir, Anish, et al. "Continuous Bayesian Model Selection for Multivariate Causal Discovery." Forty-second International Conference on Machine Learning.
>
> > Assumes training distributions generalize
>
> We agree that the effectiveness of our model depends on how well the training prior captures the target domain. This is a known limitation, and we discuss it in the paper. A conservative strategy is to use a broad prior during training to encourage generalization. Moreover, our framework is flexible: if performance in a specific application is suboptimal due to prior mismatch, the model can be easily fine-tuned on more domain-aligned priors [2].
>
> We also demonstrate empirically:
>
> - L325 (Out-of-distribution testing) — showing that, although MACE-TNP performs poorly when the prior is misspecified, we can train a MACE-TNP on a larger mix of priors to recover in-distribution performance for each prior class.
>
> - Section 5.4 — showing that MACE-TNP performs competetively with baselines on real-world data, where the true data-generating process is unknown. The prior used for this experiment is detailed in Appendix B.3.
>
> [2] Scholz, Jonas, et al. "Sim2real for environmental neural processes." arXiv preprint arXiv:2310.19932 (2023).
>
> > Potential sensitivity: As a meta-learned model, it may be sensitive to hyperparameters or require large, well-designed training datasets to achieve robust performance.
>
>
> We agree that the choice of training distribution (prior) plays a key role in performance. The choice of prior for tasks without any knowledge is an active research area [3], and future iterations of our approach could benefit from advances in identifying realistic synthetic priors.
>
> Regarding architectural hyperparameters, Appendix C.2.2 includes ablations on attention types, number of MoG components, model dimensions, and transformer depth. While some configurations perform slightly better, the differences are modest, suggesting that MACE-TNP is relatively robust to these design choices.
>
> [3] Zhang, X., & Maddix Robinson, D. (2025, July 22). Mitra: Mixed synthetic priors for enhancing tabular foundation models. Amazon Science. Retrieved from Amazon Science blog.
>
> ---
>
> We would like to thank the reviewer once again for their feedback and hope that our responses have satisfactorily addressed the raised concerns.

---

> ### Comment · Reviewer_2FPW · 2025-08-04
> **Re: authors' response**
>
> I am reasonably satisfied with the authors' response. Some of the issues raised are common in the field and largely beyond the authors' control; therefore, I will not penalize them for these.

---

### Official Review · Reviewer_nB8D · 2025-06-30

**Clarity:** 3
**Significance:** 2
**Originality:** 3
**Rating:** 4
**Confidence:** 3

**Summary:**

This paper studies the problem of evaluating interventional distributions / causal effects from the observational data without access to a detailed causal graph describing the underlying data-generating mechanisms. The authors focus on a specific family of causal models where there is no unobserved confounder. They also assume access to the Bayesian prior over the possible causal graph and parameters of the underlying causal models. Based on these knowledge, the authors propose a variational inference method to approximate the posterior distribution over the interventional distributions provided with the observational data. The simulation results support the proposed method for linear causal models and causal models with functions parametrized as Gaussian processes.

**Questions:**

1. How is the principle of meta-learning applied to this problem setting?
2. What are the parametric model assumptions that enable the proposed method?
3. In Line 72, the authors state, "Like a majority of works that learn causal models from data, we assume no hidden confounders." This statement is controversial, as unobserved confounding is one of the key challenges to causal inference. A majority of advancements in causal estimation after 2000 have focused on addressing the issues of confounding bias. For example, there is a growing line of work for identifying causal effects provided with a partial description of causal knowledge represented as a Markov equivalence class (Jiji Zhang, 08; Malinsky & Spirtes, 16; Jaber, Zhang & Bareinboim 19).

**Ethical Concerns:**

["NO or VERY MINOR ethics concerns only"]

**Final Justification:**

The authors have addressed my concerns regarding the posterior distribution inference.

**Limitations:**

The authors have not adequately addressed the theoretical limitation of their work. The model assumptions underlying the proposed method are unclear. Regarding the societal impact, the paper is mainly theoretical, and its long-term implications remain unclear.

**Paper Formatting Concerns:**

There are no major formatting issues in this paper.

**Quality:**

2

**Strengths And Weaknesses:**

This paper studies an exciting problem of causal inference without access to a complete causal graph. The direction of leveraging generative modeling is promising. Overall, this paper is well organized, and the simulation results support the proposed method under a specific family of causal models. However, I do have some questions and concerns, summarized below.

1. The relations to the meta-learning method are somewhat unclear. Typically, meta-learning algorithms refer to the methods that learn to learn. For example, a popular line of meta-learning approach (Finn, Abbeel & Levine, 2017) learns an optimal initialization point for generalizing to any future environment after taking a few steps of gradient descent. It could also be framed in terms of recurrent models that are trained via a meta-learning objective. This paper focuses on learning a posterior distribution over the potential outcomes provided with observational data. It is unclear how this learning objective relates to the existing meta-learning literature.

2. The model assumptions underlying the proposed method are unclear. The proposed method requires access to the detailed parametric knowledge about the underlying data-generating models to properly assign the prior. The authors should list these model assumptions more explicitly, similar to [57]. Otherwise, it would be confusing when compared to other non-parametric methods that attempt to leverage qualitative knowledge, e.g., (Jiji Zhang, 08).

3. The learning objective here is the posterior interventional distribution, averaging over all possible models. It would be appreciated if the authors could provide more insights on this learning objective. Since the posteior could be sensitive to the prior and one may not have access to the correct prior, it is possible to assign a prior distribution that leads to a posterior interventional distribution that is overly concentrated towards the mean. In this case, it is sufficient to learn a neural process model that recovers the first moment of the posterior but underestimates the variance. The learned posterior interventional distribution would not be able to capture the representations if the ground-truth model is an extreme case, located around the margin. This is one of the main reasons why existing causal partial identification focuses on the quantile evaluation over the boundary case (Balke & Pearl, 1994; Zhang, Tian & Bareinboi, 2021). This issue could be clarified by comparing the proposed method with existing bounding literature over a small number of candidate graphs over a discrete domain. In that case, one could clearly plot the upper and lower bound over the target causal effect, and compare how the learned posterior covers the whole feasible region. One such example is Figure 3 in (Zhang, Tian & Bareinboi, 2021).

---

> ### Author Rebuttal · Authors · 2025-07-31
>
> We thank the reviewer for their thoughtful feedback. We’re pleased that the reviewer described our paper as addressing “an exciting problem” and noted that “the direction of leveraging generative modeling is promising.” We also appreciate their positive remarks that the "paper is well organized, and the simulation results support the proposed method under a specific family of causal models".
>
> **Summary**
>
> We would like to bring some clarifications to the summary provided by the reviewer.
>
> > "… assume access to the Bayesian prior over the possible causal graph and parameters of the underlying causal models."
>
> **We do not assume access to the "true" Bayesian prior**, but rather that we can construct an empirical Bayesian prior with enough support that it is informative for the test data (whose performance is the one that is most relevant in the end).
>
> Unlike many other methods, if we have reason to believe the prior is not wide enough (e.g. due to poor prediction on a held out set), **we can easily incorporate synthetic datasets from an even wider variety of Bayesian causal models (BCM) (def. 2.1) at training time and thus widen the support**.
>
> We illustrate this in section 5.4 where we test our method on a real-life dataset, implying we do not have access to the true data generating process.
>
> > "… a variational inference method"
>
> **We do not use variational inference in the training**. For a single task (defined in L186), the task-dependent loss (Eq. 4) involves computing the log-likelihood of the interventional targets (sampled from the BCM) conditioned on the observational data. This is available in closed form. Furthermore, since tasks exhibit statistical similarity, we adopt a meta-learning approach, computing the final objective as an expectation over the task distribution.
>
> > "...functions parametrized as Gaussian processes"
>
> In our experiments, we also include neural network-based functional mechanisms. For a description of how we construct our synthetic tasks, we refer the reviewer to appendix B.2 and B.3.
>
> **Weaknesses and Questions**
>
> > How is the principle of meta-learning applied to this problem setting?
>
> Neural Processes (NPs) are models that learn to map datasets to predictive distributions, where **each dataset is sampled from a distribution over tasks that share common statistical structure**. This two-level hierarchy is captured in our loss function (Eq. 4), which reflects learning both within each task (as in standard supervised learning) and across tasks via the expectation over task distributions (where we denote a task with $\xi$) (L186) — the latter enables the model to improve its performance on unseen tasks (i.e., meta-learning or learning to learn). For a comprehensive overview of Neural Processes, we refer the reader to Section 1.2 of [1], as well as to the original papers that classify neural processes as a meta-learning paradigm [2, 3].
>
> In short, **meta-learning is happening at training time, where the loss involves an outer expectation over statistically-similar tasks**.
>
> [1] Bruinsma, W. P. (2022). Convolutional Conditional Neural Processes (PhD thesis, Department of Engineering, University of Cambridge). University of Cambridge.
>
> [2] Garnelo, Marta, et al. "Conditional neural processes." International conference on machine learning. PMLR, 2018.
>
> [3] Garnelo, Marta, et al. "Neural processes." arXiv preprint arXiv:1807.01622 (2018).
>
> > What are the parametric model assumptions that enable the proposed method?
>
> In our method, at train time, a user can specify any prior (or a mixture of priors) by specifying the Bayesian causal models (BCMs) (def. 2.1) that the training data is sampled from. At test time the posterior interventional distribution is obtained by a forward pass through the network of the test dataset. Thus, **the only model assumptions in our approach are the ones determined by the choice of BCMs we train our model on**. Our method does not rely on any specific choice of BCM, and can be changed depending on the problem.
>
> We have specified the BCMs that generated the training data (and hence determine the prior) for each experiment in Appendix B. For further clarity, we will include the description of the BCMs that generated the data for each experiment in the main text.
>
> > The learning objective here is the posterior interventional distribution, averaging over all possible models. It would be appreciated if the authors could provide more insights on this learning objective.
>
> Let us unpack the learning objective of MACE-TNP, stated in Eq. 4. As mentioned above, the objective is hierarchical:
> 1) At the task level, the quantity we want to learn is the true posterior interventional distribution. Thus, at training time, we seek to minimise the KL divergence between this quantity and the output of our model. Equivalently, this can be framed as maximising the expected log predictive distribution of the model under the interventional distribution (i.e. interventional targets conditioned on the current task $\mathbb{E}_{X_i | \xi}$). At training time, this can be evaluated in closed-form.
> 2) Because these tasks share statistical similarities, we take a meta-learning approach where we also average over multiple tasks  $\xi$ (which are defined by the observational data $\mathcal{D}_{\text{obs}}$, the interventional node $j$ and value $x_j$, and the outcome node $i$).
>
> Combined, these two lead to Eq. 4.
>
> > ... existing causal partial identification focuses on the quantile evaluation over the boundary case (Balke & Pearl, 1994; Zhang, Tian & Bareinboi, 2021).
>
> We appreciate the reviewer highlighting this point.
> The work you shared [4], and others [5], are similar in spirit to ours. In their approach, credible regions (e.g., Figure 3 in [4]) are derived from a posterior distribution obtained via MCMC, using a specified prior (see Section 3: Bayesian Approach for Partial Identification in [4]).
>
> In contrast, **our method offers an alternative way to obtain a posterior over interventions, without relying on MCMC**. As demonstrated in our experiments, our approach outperforms MCMC-based methods such as BCI-GPN.
> Samples from the posterior of the neural process can be used to estimate the (support) credible region.
> Importantly, our framework does not commit to a specific prior choice: if there are concerns about underestimating credible region widths, one can incorporate broader or even uninformative priors in the training process without modifying the core methodology.
> Our contribution is thus orthogonal to the choice of prior.
>
> [4] Zhang, Junzhe, Jin Tian, and Elias Bareinboim. "Partial counterfactual identification from observational and experimental data." International Conference on Machine Learning. PMLR, 2022.
>
> [5] Chickering, David Maxwell, and Judea Pearl. "A clinician's tool for analyzing non-compliance." Proceedings of the National Conference on Artificial Intelligence. 1996.
>
> > In Line 72, the authors state, "Like a majority of works that learn causal models from data, we assume no hidden confounders."
>
> We agree with the reviewer that the original statement was overstated and will revise it accordingly. Our assumption of causal sufficiency was made to maintain consistency with the baselines (and a lot of works on causal discovery), which also rely on this assumption. That said, we acknowledge the existence of methods that do not assume causal sufficiency.
>
> Importantly, our framework **can accommodate hidden confounding** by training on data generated from BCMs with unobserved confounders. We conducted an experiment using a simple 3-node linear model, demonstrating that MACE-TNP correctly identifies the interventional distribution in the presence of observed confounding and appropriately reflects uncertainty when a confounder is hidden (and thus the interventional distribution is not identifiable) (similar to Figure 3 in our work). Unfortunately, because we are not allowed to include any images in the rebuttal, we cannot share the results at this stage, but we will include them in the revised version.
>
> We thank the reviewer for raising this important point, as the additional experiment further highlights the flexibility of MACE-TNP in estimating interventional distributions, even under hidden confounding.
>
> ---
>
> If the reviewer feels we have addressed the majority of their concerns, we would appreciate it if they reconsidered their score. We are also happy to further address any remaining concerns.

---

> > ### Comment · Reviewer_nB8D · 2025-08-04
> >
> > I thank the authors for their response. They have addressed some of my concerns regarding the posterior inference. I will raise my score to 4.

---

### Official Review · Reviewer_RHFT · 2025-07-01

**Clarity:** 3
**Significance:** 3
**Originality:** 3
**Rating:** 4
**Confidence:** 3

**Summary:**

This paper studies an interesting problem, directly estimating the Bayesian model-averaged interventional distribution from observational data with a meta-learning framework. It addresses the major challenge of computational intractability in averaging over uncertain causal graphs and functional mechanisms, which is critical when the causal structure is not known or is only partially identified. This paper proposes MACE-TNP, a Transformer Neural Process architecture that amortizes the Bayesian inference pipeline, and demonstrates empirical convergence to the true interventional posterior under both identifiable and non-identifiable settings. This paper shows scalability to high-dimensional data (up to 40 variables) while outperforming strong Bayesian baselines.

**Questions:**

1. How does the model behave when the test data is generated from functional mechanisms not covered during training? Are there known inductive biases that improve generalization?
2. Could MACE-TNP be extended to support counterfactual queries? What modifications would be needed in the architecture or loss?
3. What is the sensitivity of the model to initialization, training set size, or hyperparameter choices? How robust is the meta-learning process?
4. Have you assessed how well-calibrated the estimated interventional distributions are? Could calibration methods be integrated during training?

**Ethical Concerns:**

["NO or VERY MINOR ethics concerns only"]

**Final Justification:**

I have carefully read the comments by the authors and other reviews by other reviewers and responses by the authors, my concerns have been addressed and I will keep my score unchanged.

**Limitations:**

yes

**Quality:**

3

**Strengths And Weaknesses:**

Strengths: 1. The paper correctly identifies a fundamental bottleneck in causal inference, making decisions based on a single inferred graph leads to overconfidence and incorrect interventions. The use of Bayesian averaging over graphs is sound and aligns with established theory. 2. Instead of traditional two-stage Bayesian inference (graph posterior + function posterior), MACE-TNP learns an end-to-end mapping from data to interventional distributions. This bypasses expensive intermediate computations and reduces approximation compounding. 3. The framework is well-grounded in Bayesian theory. Equation (3) formalizes the Bayesian model-averaged interventional distribution, and MACE-TNP aims to learn this via minimizing KL divergence over a synthetic task distribution. 4. Unlike baselines relying on restrictive assumptions (e.g., Gaussian or additive noise models), MACE-TNP handles both neural network-based and GP-based structural equations.

Weaknesses: 1. Although inference is efficient (forward pass), the training phase is expensive: requires many synthetic SCM samples, uses transformer layers with quadratic complexity, needs large-scale simulation of interventional and observational datasets. 2. The model learns from synthetic Bayesian causal models. It implicitly relies on the fidelity of the training task distribution to approximate real-world data complexity. If the real-world mechanism falls outside this distribution, performance may degrade. 3. While the model converges empirically, no theoretical guarantees are given about identifiability under finite data or architectural constraints. This may limit trust in sensitive domains (e.g., policy, medicine). 4. The model estimates interventional distributions, but it does not output interpretable causal graphs or functions, which might be desirable in scientific applications where explainability is key.

---

> ### Author Rebuttal · Authors · 2025-07-31
>
> We thank the reviewer for their thoughtful and detailed feedback. We’re glad they recognised the importance of addressing the challenge of averaging over uncertain causal structures, and appreciated our proposed end-to-end approach with MACE-TNP. We're especially pleased by the reviewer’s positive remarks on the **theoretical grounding** of our framework, its ability to **bypass costly intermediate inference steps**, its **flexibility** across different structural equation models, and its **scalability** to high-dimensional settings.
>
> **Weaknesses and Questions**
>
> > ... training phase is expensive: requires many synthetic SCM samples, ... large-scale simulation of interventional and observational datasets.
>
> We appreciate the reviewer highlighting this important point, which we also acknowledge in our limitations section (L371). While the training phase does require generating many synthetic SCM samples, this strategy is consistent with recent successful approaches in the machine learning community—such as TabPFN [1], a state-of-the-art method for tabular data that relies on large-scale synthetic training.
>
> Crucially, our method incurs a one-time training cost, after which inference becomes highly efficient, requiring only a single forward pass (as the reviewer also notes). This makes the approach particularly appealing in settings with frequent inference-time queries, where the upfront cost is effectively amortised.
>
> [1] Hollmann, N., Müller, S., Purucker, L., Krishnakumar, A., Körfer, M., Hoo, S. B., Schirrmeister, R. T., & Hutter, F. (2025). Accurate predictions on small data with a tabular foundation model. Nature, 637(8045), 319–326. https://doi.org/10.1038/s41586-024-08328-6
>
> > ... uses transformer layers with quadratic complexity
>
> While MACE-TNP currently uses full attention, the architecture is not inherently constrained to this. It can readily incorporate efficient attention mechanisms--such as linear or kernelized attention [2, 3] and low-rank approximations [4, 5]--to significantly reduce computational cost with minimal impact on performance.
>
> [2] Katharopoulos, A., Vyas, A., Pappas, N., & Fleuret, F. (2020, November). Transformers are rnns: Fast autoregressive transformers with linear attention. In International conference on machine learning (pp. 5156-5165). PMLR.
>
> [3] Choromanski, K., Likhosherstov, V., Dohan, D., Song, X., Gane, A., Sarlos, T., ... & Weller, A. (2020). Rethinking attention with performers. arXiv preprint arXiv:2009.14794.
>
> [4] Wang, S., Li, B. Z., Khabsa, M., Fang, H., & Ma, H. (2020). Linformer: Self-attention with linear complexity. arXiv preprint arXiv:2006.04768.
>
> [5] Xiong, Y., Zeng, Z., Chakraborty, R., Tan, M., Fung, G., Li, Y., & Singh, V. (2021, May). Nyströmformer: A nyström-based algorithm for approximating self-attention. In Proceedings of the AAAI conference on artificial intelligence (Vol. 35, No. 16, pp. 14138-14148).
>
> > How does the model behave when the test data is generated from functional mechanisms not covered during training?
>
> We agree this is an important limitation and explicitly acknowledge it in L373. Indeed, performance degradation when the real-world mechanism falls outside of the prior support is a general concern for all Bayesian methods.
>
> However, a key advantage of neural processes is their flexibility in incorporating a diverse mixture of priors, easily widening the support and thus improving robustness. We demonstrate this empirically in:
>
> - L325 (Out-of-distribution testing) — showing that, although MACE-TNP performs poorly when the prior is misspecified, we can train a MACE-TNP on a larger mix of priors to recover in-distribution performance for each prior class.
>
> - Section 5.4 — showing that MACE-TNP performs competitively with baselines on real-world data, where the true data-generating process is unknown (and might not be covered by our prior). The prior used for this experiment is detailed in Appendix B.3.
>
> > While the model converges empirically, no theoretical guarantees are given about identifiability under finite data or architectural constraints. This may limit trust in sensitive domains (e.g., policy, medicine)
>
> We're unsure the identifiability of which quantity you are referring to here.
> - If the underlying causal structure (and interventional distribution) is identifiable, the posterior will concentrate on the correct interventional distribution [6,7]. If not, the posterior will place mass on the interventional distributions consistent with the equivalence class of causal structures that is identifiable from observations. This is shown in Section 5.1.
> - While the recovery of the true posterior is only guaranteed in the limit of infinite tasks (L203), we do show empirically that in the finite task case we outperform baseline methods.
>
> [6] Dhir, Anish, et al. "Continuous Bayesian Model Selection for Multivariate Causal Discovery." Forty-second International Conference on Machine Learning.
>
> [7] Lungu, Valentinian, et al. "Bayesian causal discovery: Posterior concentration and optimal detection." arXiv preprint arXiv:2507.16529 (2025).
>
>
>
> > The model estimates interventional distributions, but it does not output interpretable causal graphs or functions, which might be desirable in scientific applications where explainability is key.
>
> We agree this is a valid concern—our experiments focus on accurately estimating interventional distributions, rather than recovering interpretable causal graphs or functions. However, since the model can output interventional effects for all node pairs, it is in principle possible to reconstruct the underlying causal structure from these outputs.
>
> Moreover, our framework is flexible and could be extended to jointly estimate a posterior over causal graphs—for example, by adding an auxiliary decoder head, as done in [8]. We see this as a promising direction for future work where interpretability is essential, and will mention it in the Conclusions section of the updated manuscript. We thank the reviewer for highlighting this point.
>
> [8] Dhir, Anish, et al. "A meta-learning approach to Bayesian causal discovery." 2024.
>
> >  Are there known inductive biases that improve generalization?
>
> We incorporate basic inductive biases into our architecture as discussed in the paragraph starting at L212.
> While other inductive biases such as translation equivariance [9] have been incorporated into neural processes for spatiotemporal prediction, the suitability of these additional biases is task-dependent.
>
> [9] Ashman, Matthew, et al. "Translation Equivariant Transformer Neural Processes." Forty-first International Conference on Machine Learning.
>
>
> > Could MACE-TNP be extended to support counterfactual queries? What modifications would be needed in the architecture or loss?
>
> We believe this approach holds promise for efficiently computing counterfactual queries (and their posteriors). However, pursuing this direction would require substantial modifications at both the architectural and data generation levels, likely constituting a research project in its own right.
>
> > What is the sensitivity of the model to initialization, training set size, or hyperparameter choices? How robust is the meta-learning process?
>
> In essence, MACE-TNP incorporates a transformer architecture, enabling us to draw on the extensive literature regarding effective initialization and training of such models. While we employ standard architectural choices, the method is flexible and can be adapted to any architecture the user prefers.
>
> Furthermore, we provide an ablation of a set of hyperparmeters (i.e. MoG components, attention mechanism, model size, etc.), and some resulting guidance, in Appendix C.2.2. All configurations yield reasonable performance, with slight improvements observed as the architecture size or number of MoG components increases. This suggests that the model is both robust to a range of design choices and able to benefit from scaling.
>
> The reviewer is right to highlight training set size as an important factor, particularly in the context of meta-learning. While we did not investigate the sample efficiency of our method, we agree that it is an important direction for future research. However, empirically evaluating sample efficiency can be computationally expensive.
>
> > Have you assessed how well-calibrated the estimated interventional distributions are? Could calibration methods be integrated during training?
>
> A lower score on our metric - NLPID (averaged over multiple datasets)- implies that the distribution is closer in KL to the true distribution [10].
> Hence we expect the models with a lower score to be better calibrated under the assumption that datasets have been generated from the prior.
> If this is not the case, post hoc calibration methods can indeed be used to better calibrate the model [11].
>
> [10] David Madigan and Adrian E Raftery. Model selection and accounting for model uncertainty in graphical models using occam’s window.
>
> [11] Fong, Edwin, and Chris C. Holmes. "Conformal bayesian computation." Advances in Neural Information Processing Systems 34 (2021).
>
> ---
>
> We thank the reviewer once again for their thoughtful and constructive feedback. We are happy to address any remaining questions, and if we have satisfactorily addressed the majority of the concerns, we would be grateful if the reviewer would consider revisiting their score.

---

### Official Review · Reviewer_vjb8 · 2025-07-03

**Clarity:** 3
**Significance:** 2
**Originality:** 2
**Rating:** 4
**Confidence:** 3

**Summary:**

The paper introduces MACE-TNP, a meta-learning framework designed to estimate interventional distributions under uncertain causal graphs by directly mapping observational data to Bayesian model-averaged interventional distributions using a Transformer Neural Process. Unlike traditional methods that rely on computationally expensive graph averaging or two-stage approximations, MACE-TNP leverages end-to-end training on synthetic data from Bayesian Causal Models to amortize inference. The model features variable-specific MLPs, a permutation-invariant transformer encoder, and a Mixture of Gaussians decoder, trained to minimize KL divergence from true interventional distributions. Theoretically sound and empirically validated across settings from small-scale (2–3 nodes) to large-scale (20–40 nodes) systems, MACE-TNP outperforms existing Bayesian and non-Bayesian baselines and shows promise on real-world data (e.g., Sachs dataset). Despite its high training cost and sensitivity to out-of-distribution functions, it offers a scalable and accurate solution for causal inference under uncertainty, marking a significant step forward in the field.

**Questions:**

See above weakness

**Ethical Concerns:**

["NO or VERY MINOR ethics concerns only"]

**Final Justification:**

Several of my concerns has been resolved and thus I will maintain my positive score.

**Limitations:**

Generalization may be a limitation.

**Quality:**

3

**Strengths And Weaknesses:**

## Strengths

1. **Computational Efficiency**
   - Avoids expensive two-stage Bayesian inference by amortizing computation into a single forward pass
   - Scales to moderately high-dimensional systems (up to 40 nodes)

2. **Theoretical Foundations**
   - Provably converges to true Bayesian posterior in identifiable cases
   - Properly handles both identifiable and non-identifiable graphs
   - Incorporates interventional data to resolve non-identifiability

3. **Architectural Innovations**
   - Transformer-based neural process handles permutation symmetries
   - Variable-specific embeddings distinguish intervention/outcome roles
   - Mixture of Gaussians decoder captures complex distributions

4. **Empirical Performance**
   - Outperforms Bayesian baselines (DiBS-GP, ARCO-GP) on synthetic tasks
   - Beats non-Bayesian methods (DECI, NOGAM+GP) in accuracy
   - Shows robustness to functional misspecification
   - Validated on real-world Sachs dataset

5. **Practical Design**
   - Includes ablation studies and architecture analysis
   - Discusses limitations and future improvements

## Weaknesses

1. **Fundamental Limitation: Latent Confounding**
   - Cannot handle unmeasured confounders (assumes causal sufficiency)
   - With latent variables, graph space becomes infinite
   - Theoretical results suggest this case may be undecidable as it may require the model to learn a classifier of infinite number of classes implicitly (https://arxiv.org/pdf/2411.17898)

2. **Training Requirements**
   - Needs large amounts of synthetic training data (minor)
   - Computationally expensive meta-training phase (minor)
   - Performance depends on match between training and test distributions (minor)

3. **Generalization Challenges**
   - Limited out-of-distribution robustness
   - May not extrapolate to unseen causal mechanisms

4. **Scalability Constraints**
   - Attention mechanism has quadratic complexity
   - Untested on very large graphs (>100 nodes)
   - Memory intensive for high-dimensional interventions

5. **Methodological Comparisons**
   - Lacks comparison to recent table-pfn based approaches such as causalPFN(https://arxiv.org/pdf/2506.06039)
   - Doesn't benchmark against latent variable methods
   - Limited discussion of failure modes

---

> ### Author Rebuttal · Authors · 2025-07-31
>
> We thank the reviewer for their thoughtful and constructive feedback. We’re glad they appreciated, among others, the **theoretical foundations** of our work, the ability of MACE-TNP to **avoid expensive two-stage Bayesian inference** through amortised learning, and its **strong empirical performance** across both synthetic benchmarks and real-world data.
> We appreciate their comment that our approach **"offers a scalable and accurate solution for causal inference under uncertainty, marking a significant step forward in the field"**.
>
> > Fundamental Limitation: Latent Confounding
>
> As also noted in our response to Reviewer nB8D, this is a valid concern. While we assume causal sufficiency to ensure a fair comparison with existing baselines, our method does not inherently require this assumption.
>
> In fact, MACE-TNP **can accommodate hidden confounding** by training on data generated from Bayesian Causal Models (BCMs) with unobserved confounders. To demonstrate this, we conducted an experiment using a three-node linear SCM. We trained two MACE-TNP models: one under observed confounding and one under hidden confounding. As expected, the model trained with observed confounding accurately recovered the interventional distribution, while the one trained under hidden confounding appropriately expressed higher uncertainty (similar to Figure 3), reflecting the fundamental identifiability limits.
>
> Although we're unable to include figures in the rebuttal, we will include these results in the revised manuscript to illustrate the model’s behaviour under hidden confounding, and thank the reviewer for highlighting this important point.
>
> > Training Requirements
>
> This is indeed another limitation of our approach that we acknowledge in L371. However, we argue that this strategy is consistent with recent successful approaches in the machine learning community—such as TabPFN [1], a state-of-the-art method for tabular data that relies on large-scale synthetic training.
>
> Crucially, although training requirements are high, inference-time requirements are light, requiring a single forward pass. This makes our approach particularly appealing in settings with frequent inference-time queries, where the upfront training cost is effectively amortised.
>
> [1] Hollmann, N., Müller, S., Purucker, L., Krishnakumar, A., Körfer, M., Hoo, S. B., Schirrmeister, R. T., & Hutter, F. (2025). Accurate predictions on small data with a tabular foundation model. Nature.
>
> > Generalization Challenges: Limited OOD robustness, may not extrapolate
>
> We also acknowledge this limitation in L373 and thank the reviewer for pointing it out. However, we would like to stress that
> - When tested on real-world data (Section 5.4), our approach is competitive, suggesting that it is possible to generalise to real-life data better than the considered benchmarks.
> - Robustness to prior misspecification can be improved by training on a broader mix of prior classes (see Appendix C.2.2). While selecting appropriate priors for unknown tasks is non-trivial, this is an active and emerging research area [2] which we plan to explore in future work.
>
> [2] Zhang, X., & Maddix Robinson, D. (2025, July 22). Mitra: Mixed synthetic priors for enhancing tabular foundation models. Amazon Science. Retrieved from Amazon Science blog.
>
> > Scalability Constraints: Attention mechanism has quadratic complexity
>
> While we currently use full attention, the architecture of MACE-TNP is not inherently constrained to this. It can readily incorporate efficient attention mechanisms---such as linear or kernelized attention [3, 4] and low-rank approximations [5, 6]---to significantly reduce computational cost with a relatively low impact on performance.
>
> [3] Katharopoulos, A., Vyas, A., Pappas, N., & Fleuret, F. (2020, November). Transformers are rnns: Fast autoregressive transformers with linear attention. In International conference on machine learning (pp. 5156-5165). PMLR.
>
> [4] Choromanski, K., Likhosherstov, V., Dohan, D., Song, X., Gane, A., Sarlos, T., ... & Weller, A. (2020). Rethinking attention with performers. arXiv preprint arXiv:2009.14794.
>
> [5] Wang, S., Li, B. Z., Khabsa, M., Fang, H., & Ma, H. (2020). Linformer: Self-attention with linear complexity. arXiv preprint arXiv:2006.04768.
>
> [6] Xiong, Y., Zeng, Z., Chakraborty, R., Tan, M., Fung, G., Li, Y., & Singh, V. (2021, May). Nyströmformer: A nyström-based algorithm for approximating self-attention. In Proceedings of the AAAI conference on artificial intelligence (Vol. 35, No. 16, pp. 14138-14148).
>
> > Scalability Constraints: Untested on very large graphs (>100 nodes)
>
> To address this concern, we conducted an additional experiment using the model from Sections 5.3 and 5.4, which was trained on graphs with 5–40 nodes. We evaluated it on a new test dataset consisting of 100-node graphs with approximately 400 expected edges. Importantly, the new dataset was only used at test time--**no samples from the 100-node dataset were included in training**. We obtained the NLPID for our model and for DECI, the most competitive baseline. Whereas our model has a NLPID score of $671.4 \pm 5.1$, DECI achieved a NLPID score of $691.8 \pm 5.7$. With this additional experiment we prove that
> - 1) we are able to scale the model to a higher-dimensional setting;
> - 2) when tested on this higher-dimensional setting, our model continues to outperform the strongest benchmark, even if this **testing scenario is out-of-distribution for our model**.
>
> (We used one 24GB memory GPU for this experiment).
>
> > Methodological Comparisons
> > a) Lacks comparison to recent table-pfn based approaches such as causalPFN (https://arxiv.org/pdf/2506.06039)
> > b) Doesn't benchmark against latent variable methods
>
> We thank the reviewer for highlighting do-PFN, which is indeed a closely related contribution. Its appearance reinforces the growing interest in addressing interventional queries / causal effect estimation using meta-learning approaches such as Neural Processes and Prior-Data Fitted Networks.
>
> As do-PFN was released on arXiv **after the NeurIPS submission deadline**, we did not include it in our comparisons. According to NeurIPS policy, comparisons to works appearing after March 1, 2025, are not required (see the NeurIPS-FAQ).
>
> While do-PFN offers an accessible interface, it is currently limited to binary treatments in the context of CATE, and cannot be directly applied to our broader setting involving continuous treatments and general interventional distributions. Conceptually, both methods aim to learn the same target quantity but differ in architectural choices (e.g., treatment/covariate encoding, model scale, decoder parameterization (MoG in MACE-TNP vs. categorical distribution in do-PFN, etc.).
>
> We appreciate the connection and will discuss it in the revised version.
>
> > c) Limited discussion of failure modes
>
> We appreciate this point. In Lines 372–373, we do acknowledge key limitations, including out-of-distribution generalization, training cost, and scalability. Additionally, we plan to include the hidden confounder experiment in the revision to directly address the limitation related to latent confounding. If there are any other failure modes the reviewer thinks we have not properly addressed, we are happy to discuss them.
>
> ---
>
> We would like to thank the reviewer once again for their insightful feedback and, if we sufficiently addressed your concerns, we would appreciate it if you could reconsider your score.

---

### Note · Authors · 2025-08-12

**Dear SACs, ACs, and reviewers,**

We thank the reviewers for their insightful feedback and constructive suggestions, which will help us further improve the manuscript. While we acknowledge certain limitations—limitations that, as reviewer 2FPW noted, are “common in the field and largely beyond the authors’ control”—we believe our work remains a valuable contribution. As reviewer vjb8 observed, it offers “a scalable and accurate solution for causal inference under uncertainty, marking a significant step forward in the field.”

Two reviewers recognised the value of our work, either by raising their scores or indicating that our responses were satisfactory. We were unfortunately unable to engage in a similar exchange with the remaining two reviewers, as no further feedback was provided after our rebuttal. Nonetheless, we addressed their initial comments comprehensively and conducted additional experiments targeting common concerns:

- **Scalability:** To address reviewer vjb8’s point, we tested MACE-TNP on graphs with 100 nodes. Without retraining, the model outperformed the best baseline (DECI), demonstrating both scalability and strong generalisation.

- **Latent confounding:** In line with concerns from reviewers vjb8 and nB8d, we will add a 3-node linear model experiment showing that MACE-TNP correctly identifies interventional distributions under observed confounding and appropriately reflects uncertainty when confounders are hidden.

- **Interpretability:** Following suggestions from reviewers RHFT and 2FPW, we propose methods to enhance interpretability for settings, such as scientific applications, where users are also interested in the underlying causal structure.

- **Sensitivity to training data distribution:** While performance inevitably depends on training distribution—a common challenge in the field—MACE-TNP outperforms baselines on real-world data without access to the true prior. Moreover, its design allows for easily updating priors when needed.

We hope these clarifications and results address the majority of concerns and once again thank the reviewers for their thoughtful input.

**Best regards,**

The authors of Submission 24867

---

### Decision · Program_Chairs · 2025-09-17

**Decision:**

Accept (poster)

**Comment:**

This work proposes a meta-learning and model-averaging approach to infer interventional distributions. Despite limitations (expensive training, scalability, unobserved confounding,...) reviewers overall agree that the work is valuable and notably emphasize the principled approach and good experimental results with strong baselines.